# Innovative Cloud Quantification: Deep Learning Classification and Finite Sector Clustering for Ground-Based All Sky Imaging

Jingxuan Luo [1, 2], Yubing Pan [3], Debin Su [1], Jinhua Zhong [1], Lingxiao Wu [4], Wei Zhao [2],
Xiaoru Hu [1, 2], Zhengchao Qi [1, 2], Daren Lu [2] and Yinan Wang [2]

[1]College of Atmospheric Sounding, Chengdu University of Information Technology, Chengdu 610225, China;
[2]Key Laboratory of Middle Atmosphere and Global Environment Observation, Institute of Atmospheric Physics, Chinese Academy of Sciences, Beijing 100029, China
[3]Institute of Urban Meteorology, CMA, Beijing, 100089, China
[4]Key Laboratory for Cosmic Rays of the Ministry of Education, Tibet University, Lhasa 850000, China

Correspondence to: Yinan Wang (wangyinan@mail.iap.ac.cn)

**Abstract.** Accurate cloud quantification is essential in climate change research. In this work, we construct an automated computer vision framework by synergistically incorporating deep neural networks and finite sector clustering to achieve robust whole sky image-based cloud classification, adaptive segmentation, and recognition under intricate illumination dynamics. A bespoke YOLOv8 architecture attains over 95% categorical precision across four archetypal cloud varieties curated from extensive annual observations (2020) at a Tibetan highland station. Tailor-made segmentation strategies adapted to distinct cloud configurations, allied with illumination-invariant image enhancement algorithms, effectively eliminate solar interference and substantially boost quantitative performance even in illumination-adverse analysis scenarios. Compared with the traditional threshold analysis method, the cloud quantification accuracy calculated within the framework of this paper is significantly improved. Collectively, the methodological innovations provide an advanced solution to markedly escalate cloud quantification precision levels imperative for climate change research, while offering a paradigm for cloud analytics transferable to various meteorological stations.

## 1 Introduction

Clouds play a crucial regulatory role in the Earth's climate system (Voigt et al., 2021). Serving as important barriers that regulate the Earth's energy balance on a global scale, cloud layers help prevent surface overheating. Moreover, due to their reflective, absorptive, and emissive properties of solar radiation, clouds also contribute to a notable net cooling effect, playing an indispensable role in regulating the overall temperature of the Earth (Raghuraman et al., 2019). It is noteworthy that in recent years, the critical role of clouds in the Earth's radiation balance has been further emphasized and empirically demonstrated (Gouveia et al., 2017). For instance, Zhao et al. delve into detail in their latest review on how cloud layers impact the global climate system through radiation forcing mechanisms. They reveal how clouds function as a

dynamic feedback system, capable of both cooling the Earth by obstructing solar shortwave radiation and warming it by absorbing and re-emitting longwave radiation, thus exerting a significant influence on the global radiation balance (Zhao et al., 2023). Cloud quantification is the precise analysis of sky images to transform cloud body characteristics into a series of quantifiable parameters, including but not limited to cloud amount and cloud type, which are essential for understanding and modeling the Earth's radiation balance, energy transport, and climate change. However, simultaneously, the influence of clouds on the climate system varies depending on their type and altitude. For instance, high-altitude cirrus clouds, due to their strong absorption and re-emission characteristics of longwave radiation, effectively contribute to the warming (greenhouse) effect on the Earth's radiation balance. Conversely, low-level stratocumulus and cumulus typically exhibit a cooling effect due to their effective reflection and shielding of solar shortwave radiation (Werner et al., 2013). Accurately determining the type, distribution, and evolution of clouds is crucial for the long-term monitoring and prediction of climate change (Riihimaki et al., 2021). However, there are significant differences in cloud cover between different locations, and regional climate characteristics vary noticeably. Globally, cloud frequency is higher over the ocean than over land, but the situation is reversed for cloud systems with more than two layers. The seasonal variation in the global average total cloud fraction is small, but there are significant variations between different latitudinal zones (Chi et al., 2024). Precise cloud identification can provide crucial information on climate change from multiple perspectives (Jafariserajehlou et al., 2019). Additionally, it can validate the accuracy of climate model predictions and provide input parameters for climate sensitivity studies (Hutchison et al., 2019). Therefore, conducting precise cloud quantification observations is of great significance for climate change scientific research, which is precisely the starting point of this study using image processing techniques to achieve accurate cloud calculations.

Currently, accurate cloud typing and quantification still face certain difficulties and limitations. For cloud classification, common approaches include manual identification, threshold segmentation, texture feature extraction, satellite remote sensing, ground-based cloud radar detection, aircraft sounding observations, etc. (Li et al., 2017; He et al., 2018; Ma et al., 2021; Rumi et al., 2015; Wu et al., 2021). Manual visual identification relies on the experience of professional meteorological observers to discern cloud shapes, colors, boundaries and other features to categorize cloud types. This method has long been widely used, but is heavily impacted by individual differences and lacks consistency, with low efficiency (Alonso-Montesinos, 2020). Threshold segmentation sets thresholds based on RGB values, brightness and other parameters in images to extract pixel features corresponding to different cloud types for classification. It is susceptible to illumination conditions and ineffective at distinguishing transitional cloud zones (Nakajima et al., 2011). Texture feature analysis utilizes measurements of roughness, contrast, directionality and other metrics to perform multi-feature combined identification of various clouds, but adapts poorly to both tenuous and thick clouds (Yu et al., 2013). Satellite remote sensing discerns cloud types based on spectral features in different bands combined with temperature inversion results, but has low resolution and inaccurate recognition of ground-level small clouds (Yang et al., 2007). Ground-based cloud radar

differentiation of water and ice clouds relies on measured Doppler velocity and other parameters, with inadequate detection of high thin clouds (Irbah et al., 2023). Aircraft sounding observations synthesize multiple parameters to make judgments, but have limited coverage and observation time.

In the fields of meteorology and remote sensing, cloud detection and recognition have always been at the forefront and a challenge of research. Currently, the mainstream ground-based cloud detection methods primarily consist of two categories: traditional image processing techniques and deep learning-based techniques (Hensel et al., 2021). The advantages of traditional image processing techniques are mainly reflected in the easy operation and low computational cost, which are suitable for rapid preliminary

identification of cloud cover areas, however, the high sensitivity of such methods to changes in lighting conditions leads to unstable identification results under complex lighting dynamics, especially in the identification of high-altitude thin cirrus clouds, complex boundary cloud bodies, and multiple clouds, due to the lack of adaptive ability and accurate feature expression, it is difficult to achieve the ideal quantization accuracy and weak adaptability to atypical cloud types, which affects the accuracy of cloud calculation.

Deep learning methods can efficiently and accurately classify and segment cloud images under complex cloud types and various lighting conditions by means of a deep neural network model driven by large-scale training data, and significantly improve the quantization performance under unfavorable lighting environments by combining with algorithms such as image enhancement. Deep learning methods also have obvious shortcomings, such as relying on a large amount of labeled data, high-performance computational

resources, and the recognition performance in extreme lighting scenarios such as extremely bright or dark still needs to be improved. Current mainstream cloud detection methods include LiDAR measurements, satellite remote sensing inversion, ground-based cloud radar, and all-sky image recognition (Li et al., 2022a). Laser radar, by emitting sequential pulses of laser beams and deducing cloud vertical structure and optical thickness based on echo information, can directly quantify cloud amounts. There are compact or

even portable laser radar devices available on the market. However, in the context of the cloud image recognition method addressed in this study, these devices incur high costs and offer limited coverage. Satellite remote sensing inversion utilizes parameters like cloud top temperature and optical depth, combined with inversion algorithms to obtain cloud amount distribution. However, restricted by resolution, it has poor recognition of local clouds (Rumi et al., 2015). Ground-based cloud radar can measure

backscatter signals at different altitudes to determine layered cloud distribution, but has weak return signals for high thin clouds, resulting in inadequate detection. With multiple cloud layers, it struggles to differentiate between levels, unfavorable for accurate quantification (Van De Poll et al., 2006). The conventional whole sky image segmentation utilizes fisheye cameras installed at ground stations to acquire whole sky images, then segments the images based on color thresholds or texture features to calculate pixel

proportions of various cloud types, which are converted to cloud cover. This method has the advantage of easy and economical image acquisition, but is susceptible to illumination changes that can impact segmentation outcomes, with poor recognition of small or high clouds (Alonso-Montesinos, 2020). In summary, the current technical means for cloud classification and quantification lack high accuracy, cannot

precisely calculate regional cloud information, and need improved stability and reliability. They fall short of meeting the climate change science demand for massive fine-grained cloud datasets.

In recent years, with advances in computer vision and machine learning theories, some more sophisticated technical means have been introduced into cloud classification and recognition, making significant progress. While traditional methods are not able to characterize and extract cloud texture features well, convolutional neural networks can learn increasingly complex patterns and discriminative textures from large pre-trained datasets. In addition, convolutional neural networks typically employ a hierarchical feature extraction framework that captures fine textures such as edges and shapes. For instance, cloud image classification algorithms based on deep learning have become a research hotspot. Deep learning can automatically learn feature representations from complex data and construct models to synthetically judge the visual information of cloud shapes, boundaries, textures, etc. to distinguish between different cloud types (Yu et al., 2020). Meanwhile, unsupervised learning methods like k-means clustering are also widely applied in cloud segmentation and recognition. This algorithm can autonomously discover inherent data category structures without manual annotation, enabling cloud image partitioning and greatly simplifying the workflow, Krauz and other research teams have previously successfully analyzed all-sky images using the k-means clustering algorithm to quickly and efficiently delineate cloud cover and clear sky regions, significantly improving the speed and efficiency of cloud quantification tasks (Krauz et al., 2020). It can be foreseeable that the combination of deep learning and unsupervised clustering for cloud recognition will find expanded applications in meteorology. We also hope to lay the groundwork for revealing circulation characteristics, radiative effects and climate impacts of different cloud types through this cutting-edge detection approach.

Currently, many cloud recognition algorithms face significant challenges in dealing with different cloud types, especially high-altitude thin cirrus and transitional hybrid clouds (Ma et al., 2021). Among them, the traditional NRBR (Normalized Red/Blue Ratio) identification method, although able to provide preliminary cloud estimation in general, shows obvious limitations in terms of shadowing effects and identification of thin cirrus edges due to the fact that it relies only on color features to make judgments, and the variation of illumination conditions greatly affects the identification results. To address these issues and limitations, we propose constructing an end-to-end cloud recognition framework, with a focus on achieving accurate classification of cirrus, clear sky, cumulus and stratus, paying particular attention to the traditionally challenging cirrus. Building upon the categorization, we design adaptive dehazing algorithms and k-means clustering finite sector segmentation to enhance recognition of cloud edges and tenuous regions. We hope that through optimized framework design, long-standing issues of cloud typing and fine-grained quantification can be solved, significantly improving ground-based cloud detection and quantification for solid data support in related climate studies. The structure of this paper is as follows. Section 2 introduces the study area, data acquisition, and construction of the cloud classification dataset. Section 3 elaborates the methodologies including neural networks, image enhancement, adaptive processing algorithms, and evaluation metrics. Finally, Sections 4-6 present the results, discussions, and conclusions respectively.

## 2 Study Area and Data

### 2.1 Study Area

The Yangbajing Total Atmosphere Observatory (90°33′E,30°05′N) is located next to the Qinghai-Tibet Highway and Qinghai-Tibet Railway, 90 kilometers northwest of Lhasa, Tibet, in an area with an average elevation of 4,300 meters. This region has high atmospheric transparency and abundant sunlight, creating a unique meteorological environment. The Yangbajing area is far away from industries and cities, and the air quality is relatively good, which can reduce the impact of atmospheric pollution on cloud observation (Krüger et al., 2004). Meanwhile, Tibet spans diverse meteorological types, meaning various cloud types can be observed in the same area, enabling better research on the evolution patterns of different cloud types.

### 2.2 Imager Information

The cloud quantification automated observation instrument used in this study is installed at the Yangbajing Comprehensive Atmospheric Observatory (90°33'E, 30°05'N) and has been measuring since April 2019. The visible light imaging subsystem mainly comprises the visible light imaging unit (Figure 1a), the sun tracking unit (Figure 1b), the acquisition box, and the power box. As summarized in Table 1, this system images the entire sky every 10 minutes, measuring clouds ranging from 0 to 10 km with elevation angles above 15°. It can capture RGB images in the visible spectrum at a resolution of $4288 \times 2848$ pixels. This visual imaging device is equipped with a complementary metal oxide semiconductor (CMOS) image sensor system with an ultra wide angle fisheye lens design, which can regularly capture visible light spectrum images across the entire sky range; The integrated sun tracking system can accurately calculate and track the position of the sun in real-time, ensuring effective blocking of direct sunlight shining into the CMOS system, thereby protecting its sensitive photosensitive components from damage and significantly reducing the interference effect of white light around the sun on subsequent image processing.

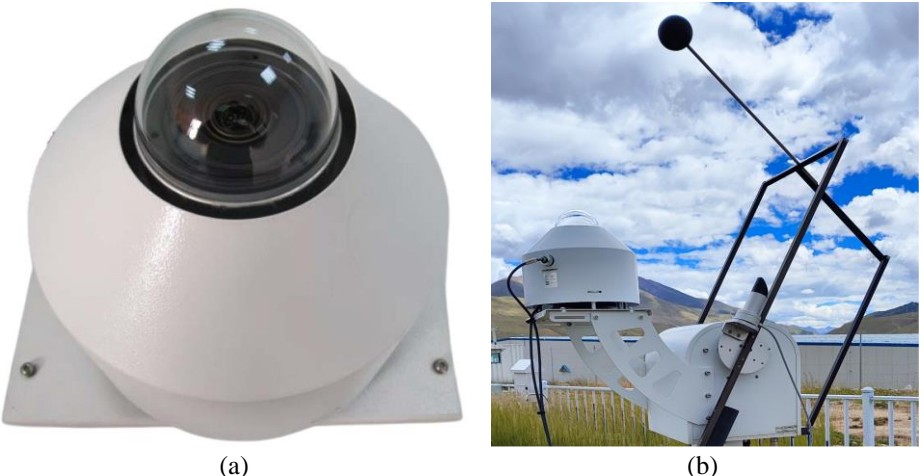

(a)                                                    (b)

**Figure 1. Automatic cloud observer. (a) Visible light imaging unit;(b) Sun tracking structure.**

**Table 1. Detailed specifications of automatic cloud observer.**

| Function | Description |
|---|---|
| Measurable cloud distance | 0~10Km |
| Measuring range | Elevation angle above 15° |
| Observation periods | Observe every 10 minutes |
| Horizontal visibility | ≥2km |
| Operating temperature | -40°~50° |
| Sensor | CMOS |
| Image resolution | 4288 × 2848 |
| Operational durability | 24 h operation |
| Ingress protection | IP65 |

## 2.3 Dataset

This study uses an all-sky image dataset between 2019 and 2022. Considering that images during sunrise and sunset hours are susceptible to lighting conditions, we only select images between 9am and 16pm hours each day. Also, to reduce the correlation, only one image is selected every half hour, which results in 15 sample images per day. Among all the selected images, rain and snow as well as obscured or contaminated lenses were excluded. Eventually, 4000 high-quality all-sky images without rain, snow, or occlusion were selected from these images and classified into four categories of 1000 images each, which were: cirrus, clear sky, cumulus, and stratus; it is important to emphasize that the division of clouds into the four main types here is intended to accurately quantify the proportion of clouds in each category, rather than considering mixed clouds. These four cloud types play an important role in the weather of the region and are the main reference factors for this classification, each type of cloud has unique visual and morphological characteristics and is fully representative of the region(Lohmann and Neubauer, 2018).

## 3 Materials and Methods

The framework proposed in this study is illustrated in Figure 2. It can be summarized into the following steps:(1) Data quality control and preprocessing. First, quality control is performed on the collected raw all-sky images to remove distorted images caused by occlusion or sensor issues. Then, image size and resolution are standardized.(2) Deep neural network classification and evaluation metrics. The YOLOv8 deep neural network is utilized to categorize the cloud images, judging which of the four types (cirrus, clear sky, cumulus, and stratus) each image belongs to. Precision, recall, and F1-scores are used to evaluate the classification performance.(3) Adaptive enhancement. Different image enhancement strategies are adopted according to cloud type to selectively perform operations like dehazing, contrast adjustment etc. to improve image quality. (4) K-means Clustering with Finite Sector Segmentation: The improved photos are subjected to category-based K-means clustering, which is based on finite sector segmentation, in order to extract cloud features and produce precise cloud detection outcomes.

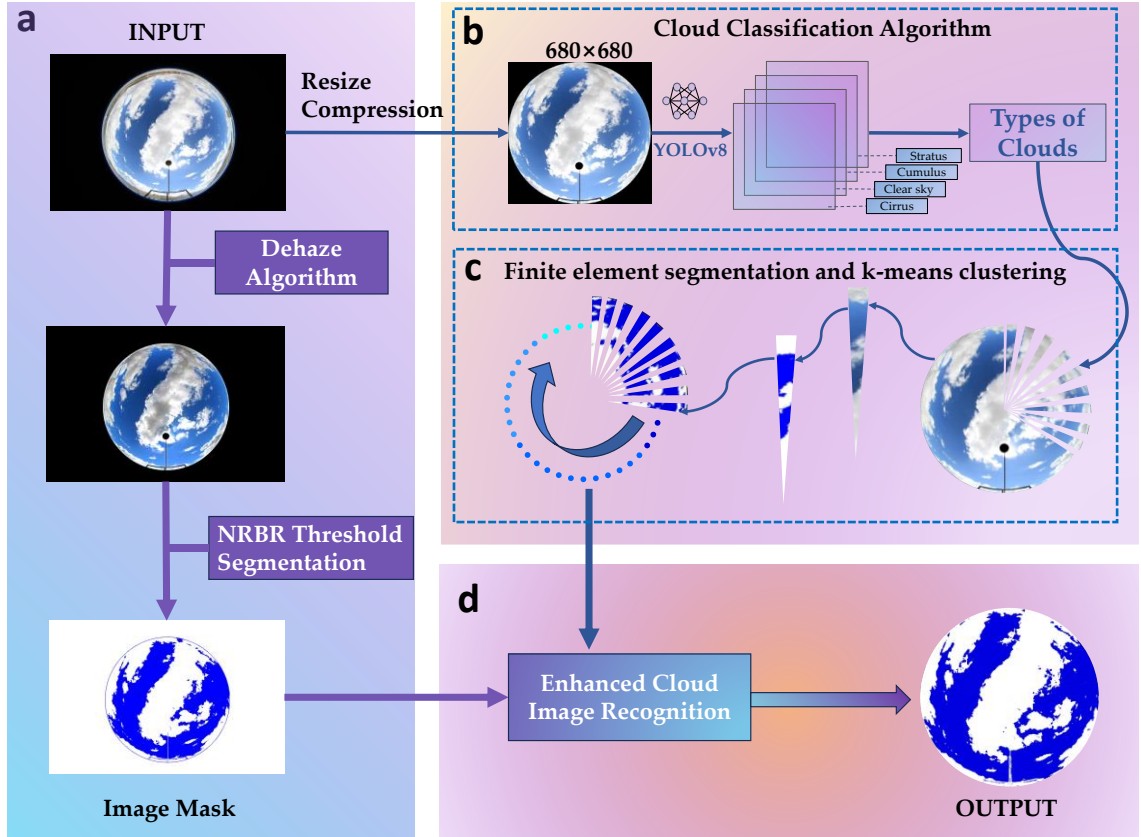

**Figure 2. Cloud detection flowchart. (a) Traditional NRBR threshold segmentation to compute the cloud amount; (b) YOLOv8 model to identify the cloud type; (c) Finite sector segmentation k-mean clustering process; (d) Local refinement for cloud identification.**

### 3.1 Quality Control and Preprocessing

Considering that irrelevant ground objects may occlude the edge areas of the original all-sky images, directly using the raw images to train models could allow unrelated ground targets to interfere with the learning of cloud features, reducing the model's ability to recognize cloud regions (Wu et al., 2023). Therefore, we cropped the edges of the original images, using the geometric center of the all-sky images as the circle center and calculating the circular coverage range corresponding to a 26° zenith angle, to precisely clip out this circular image area and remove ground objects on the edges. This cropping operation eliminated ground objects from the original images that could negatively impact cloud classification, resulting in circular image regions containing only sky elements. To facilitate subsequent image processing operations while ensuring image detail features, the cropped images underwent size adjustment to set the target resolution to 680×680 pixels. Compared to the original 4288×2848 pixels, adjusting the resolution retained the main detail features of the cloud areas in the images, but significantly reduced the file size for easier loading and calculation during network training. Finally, a standardized dataset was constructed by cloud type - the resolution-adjusted images were organized and divided into four folders for cirrus, clear sky, cumulus, and stratus, with 1000 pre-processed images in each folder. A standardized all-sky image dataset containing diverse cloud morphologies was built.

### 3.2. Deep Neural Network Classification

### 3.2.1. Network Structure Design

The main reason why YOLOv8 is the preferred framework in this study is its unique design that can effectively handle the task of all-sky image cloud classification under complex lighting conditions. Compared with the previous YOLO series and some other classic image recognition models, YOLOv8 is able to extract richer gradient flow information by adopting Darknet-53 as the Backbone and replacing the original C3 module with the improved C2f module in the Neck part. (Li et al., 2023), which is conducive to capturing the cloud's delicate textural and boundary features. Meanwhile, the PAN-FPN structure of YOLOv8 achieves model lightweighting while retaining the original high-performance performance, while the detection head part adopts a decoupled structure, which is responsible for the classification and regression tasks, respectively (Xiao et al., 2023), and adopts the binary cross-entropy loss (BCE Loss) for the optimization of the classification task, together with the distributed focus loss (DFL) and the complete IoU loss (CIoU) for bounding box regression prediction, this detection structure can significantly improve the detection accuracy and convergence speed of the model (Wang et al., 2023). Considering the limited size of the cloud dataset, we loaded the YOLOv8-X-cls model pre-trained on the ImageNet dataset as the initialization model, with a parameter count of 57.4 M. After careful module design, pre-trained model initialization, and training parameter configurations, we constructed an end-to-end cloud classification network with excellent performance.

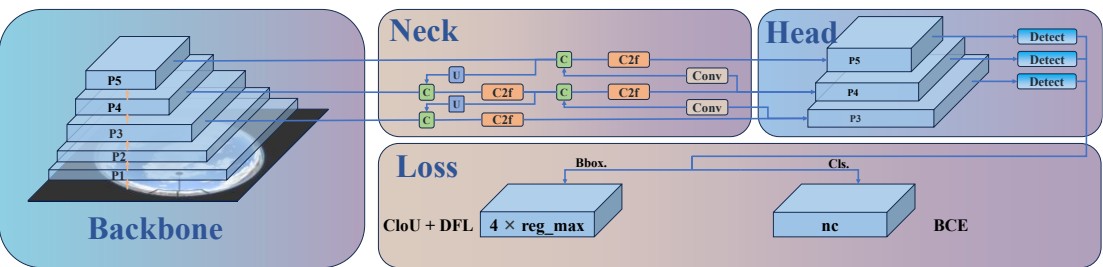

**Figure 3. YOLOv8 machine learning architecture, divided into four parts: Backbone, Neck, Head and Loss.**

### 3.2.2. Experimental Parameter Settings

After constructing the model architecture, we trained the model using the previously prepared classification dataset containing images of multiple cloud types. During training, the input image size was set to 680×680. We set the maximum number of training epochs to 400, and the number of samples used per iteration was 32. To prevent overfitting, momentum and weight decay terms were added to the optimizer and the patience parameter was adjusted to 50. To augment the sample space, various data augmentation techniques were employed such as random horizontal flipping (probability of 0.5) and mosaic (probability of 1.0). The SGD optimizer was chosen since its stochastic sampling and parameter update provide opportunities to jump out of local optima, helping locate the global optimum in a wider region. Considering initial and final learning rates, the initial learning rate was set to 0.01 and gradually decayed during training to enable more refined optimization of model parameters during later convergence.

### 3.2.3. Cloud Classification Evaluation Indicators

To comprehensively evaluate the cloud classification performance of the model, a combined qualitative and
quantitative analysis scheme was adopted. Qualitatively, we inspected the model's ability in categorizing
different cloud types, boundaries, and detail structures by comparing classification recognition differences
between the validation set and test set. Quantitatively, metrics including precision, recall and F1-score were
used to assess the model(Dev et al., 2017; Guo et al., 2024). Precision reflects the portion of true positive
cases among samples predicted as positive, and is calculated as:

$$\text{Precision} = \frac{TP}{TP+FP} \tag{1}$$

Recall represents the fraction of correctly classified positive examples out of all positive samples, and is
calculated as:

$$\text{Recall} = \frac{TP}{TP+FN} \tag{2}$$

F1-score considers both precision and recall via the formula:

$$F1 = 2 \times \frac{\text{Precision} \times \text{Recall}}{\text{Precision} + \text{Recall}} \tag{3}$$

In the above equation True Positive (TP) denotes the actual number of positive samples that the model
correctly predicts as positive category (i.e. cloud category), which represents the number of real cloud
images that the model successfully recognizes. False Positive (FP) denotes the number of samples that the
model incorrectly predicts as positive category but actually belongs to the negative category (non-cloud
category), which implies the number of cloud images that the model misidentifies. False Negative (FN)
denotes the number of samples that the model incorrectly predicted as a negative category but actually
belonged to a positive category, which represents the number of cloud images that the model failed to
identify. With this combined qualitative and quantitative evaluation system, we can comprehensively
examine the cloud classification recognition performance of the model.

### 3.3. Adaptive Enhancement Algorithm

When processing all-sky images, we face the challenges of visual blurring and low contrast caused by
overexposure and haze interference. To address this, a dark channel prior algorithm is adopted in this study.
The core idea of the dark channel prior algorithm is to perform haze estimation and elimination based on
dark channel images (Kaiming et al., 2009). First, for each pixel of the input image, the dark channel image
is computed by selecting the minimum value among its three RGB color channels. The non-zero minima in
the dark channel image are utilized to estimate the global atmospheric light intensity A. The atmospheric
light is the background light source that affects the brightness of the whole scene, and it plays a key role in
the haze scattering model. Based on the atmospheric scattering model, we can calculate the transmittance t
for each pixel point in the image, and the value indicates the visibility of the pixel point,apply the formula:

$$J(x)=I(x)-\frac{A}{t}+A \qquad (4)$$

Where J denotes the image after defogging and I is the original input image. Through the de-fogging enhancement algorithm, the fog component in the image can be effectively eliminated, making the cloud and blue sky boundary more distinct, which is conducive to the subsequent generation of high-quality cloud coverage data.

In image enhancement algorithms, the atmospheric light value A directly affects the intensity of defogging. Thanks to the powerful cloud classification network, we design an adaptive enhancement strategy after recognizing different cloud types. For thin cirrus, if the intensity is too strong, it may be filtered out, so we choose a smaller A value to retain the details; while for sunny, cumulus and stratocumulus, which are thicker, we can choose a larger A value to enhance the de-fogging effect, remove the overexposed regions

near the sun and at the edges, and obtain a more uniform sky distribution. We focus on analyzing the processing effect of two types of error-prone regions, firstly, the region around the sun is often misjudged due to overexposure, and secondly, the white light at the edge of the sky; in order to improve the segmentation quality, we adopt additional processing for these two regions, for the region around the sun, the algorithm can judge the sun position by identifying the position of the mask and thus the sun position,

and the enhanced defogging algorithm is applied to the circular region to achieve the reduction of white light For the sky edge region, after eliminating the edge features, we design a circular region at the edge of the sky, and use the enhanced defogging algorithm for this region to reduce the effect of white light on the recognition. After the above optimization design, the misjudgment problem around the sun and the edge is effectively controlled, and the cloud segmentation quality is improved.

**3.4. Finite Sector Segmentation and K-means Clustering**

Based on the cloud type classification results obtained, we propose an adaptive image segmentation method for cloud morphology as shown in Figure 4. Different cloud types exhibit different shapes and require customized segmentation strategies to get the best results. We use an adaptive image segmentation method based on cloud types to evenly divide the circular region into multiple sectors by taking the geometric

center of the full-sky image as the center of the circle, and the distance from the center to the edge of the circular sky as the radius of the circle to meet the characteristics of different cloud shapes. Cirrus are difficult to identify due to their weak shape and similar color to the sky. In order to capture the cirrus features more finely, we segment the all-sky image in which they are located into 72 sectors, and more sectors help to extract more subtle color and texture variations, which enhances the accuracy of the

clustering algorithm in distinguishing cirrus from other celestial elements. The clear sky is divided into 4 sectors to satisfy the need for effective differentiation due to the small number of elements in the image, which also avoids unnecessary subdivision, reduces computational complexity, and improves the algorithm's execution efficiency and classification accuracy in simple scenarios. Cumulus possess obvious edges, but may cause visual interference due to uneven illumination. In order to balance the capture of edge

information and the consistent processing of the internal structure, we divide it into 36 sectors, which ensures the recognition of the cloud boundary and adapts to the possible lighting differences inside the cumulus. The constituent elements in the layer cloud image are relatively few and uniform, so the same division into 4 sectors can satisfy the requirements of cluster analysis, which retains the necessary spatial resolution and avoids noise and redundant computation due to too many sectors. This adaptive segmentation strategy is based on the understanding of the four types of cloud morphological features and determined by a large number of actual test results, which significantly improves the accuracy of the clustering algorithm in recognizing the amount of clouds.

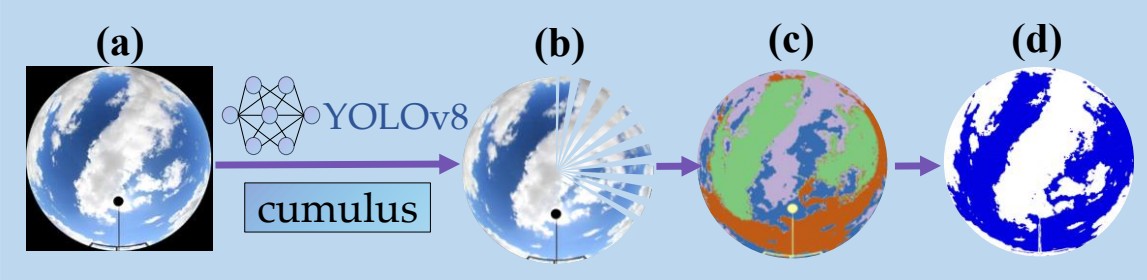

**Figure 4. Adaptive image segmentation process. (a) Image after preprocessing; (b) Sector segmentation based on cloud type; (c) Sector K-means clustering recognition; (d) Cloud recognition result.**

Upon obtaining the images segmented adaptively by cloud type, we conducted multiple experiments to determine the optimal value of K for K-means clustering within each sectorial region. The specific selection process is outlined as follows: (1) Initial setting of K values based on the complexity of the observational data and the expected number of clustering categories (such as sky, clouds, and background). (2) Implementation of the K-means algorithm and observation of clustering results. Adjustment of K values based on the actual clustering effect until the clustering results stabilize, i.e., the clustering centers no longer exhibit significant changes between adjacent iterations (Dinc et al., 2022). (3) Evaluation of clustering results under different K values using clustering validity indices such as silhouette coefficient, Calinski-Harabasz index, and Davies-Bouldin index. Selection of the K value that optimizes the evaluation indices. (4) Rationality check of the selected K value by combining meteorological expertise and practical experience to ensure consistency with meteorological principles and actual observation conditions. In this study, for the task of quantifying and classifying cloud amounts in the Yangbajing area's full-sky images, we chose K = 5 as the hyperparameter for the clustering algorithm. This decision was reached based on a series of rigorous experimental analyses and practical effectiveness evaluations. Through extensive trial and error and cross-validation with a large sample dataset, we found that when K is set to 5, the clustering results can effectively distinguish between clear blue skies, white cloud layers, transitional zones, and potential ground or near-ground obstructions, thereby achieving the desired segmentation effect. We also drew upon prior knowledge in the field regarding cloud amount and cloud feature recognition and combined it with on-site observational data to ensure that the selected K value aligns with actual physical phenomena. Given the complex and varied lighting conditions in the Yangbajing area, this clustering strategy maintains high robustness and identification efficiency under various lighting dynamics.

In traditional cloud segmentation, the Normalized Red/Blue Ratio (NRBR) threshold segmentation method exhibits certain shortcomings. Firstly, it struggles to effectively distinguish intense white light around the sun, often misclassifying these overexposed areas as cloud regions. Secondly, it fails to properly handle the bottom of thick cloud layers, where the regions appear dark due to the lack of penetrating light and may be erroneously classified as clear sky areas. Both misclassifications stem from the NRBR threshold segmentation method overly relying on RGB color features without comprehensive consideration of lighting conditions. When atypical lighting distributions occur, accurate cloud and sky differentiation becomes challenging based solely on red/blue ratio values. Therefore, after obtaining the initial cloud segmentation results, we propose a mask-based refined segmentation method to further enhance the effectiveness. The specific approach involves first extracting the predicted sky regions from the aforementioned segmentation results, using them as a mask template. Subsequently, each sector undergoes k-means clustering to identify blue sky and white clouds, restricting the region after concatenating sectors within the mask-defined blue sky template. This process yields more nuanced identification results. By conducting secondary segmentation only on key areas and leveraging the results from adaptive k-means extraction, a finer segmentation is achieved. Ultimately, building upon the initial segmentation, this approach significantly improves potential misclassifications at the cloud edges, generating more accurate final cloud detection results. This design, guided by prior masks for localized refinement, effectively enhances the quality of cloud segmentation.

**4 Results**

**4.1. Cloud Classification Results**

This study constructs a dataset based on four dominant types of cloud images collected from the Yangbaqing station in Tibet and employs the YOLOv8 deep learning model for cloud classification. To quantitatively assess the training effectiveness of the YOLOv8 cloud classification model, we record the values of the loss function and training accuracy at different training epochs, as depicted in Figures 5a. With the increase in training iterations, the model's loss value consistently decreases, with the training set loss decreasing from around 0.4 to near 0. The model gradually achieves improved predictive performance, reducing the gap between predicted values and true labels. Simultaneously, we analyze the classification accuracy curve during the model training process. As seen in Figures 5b, the model's Top-1 Accuracy rises from 0.5 to around 0.98. Through continuous training optimization, the model demonstrates sustained improvement in accuracy for distinguishing the four cloud types, progressively acquiring the ability to effectively discriminate the visual features of different cloud formations.

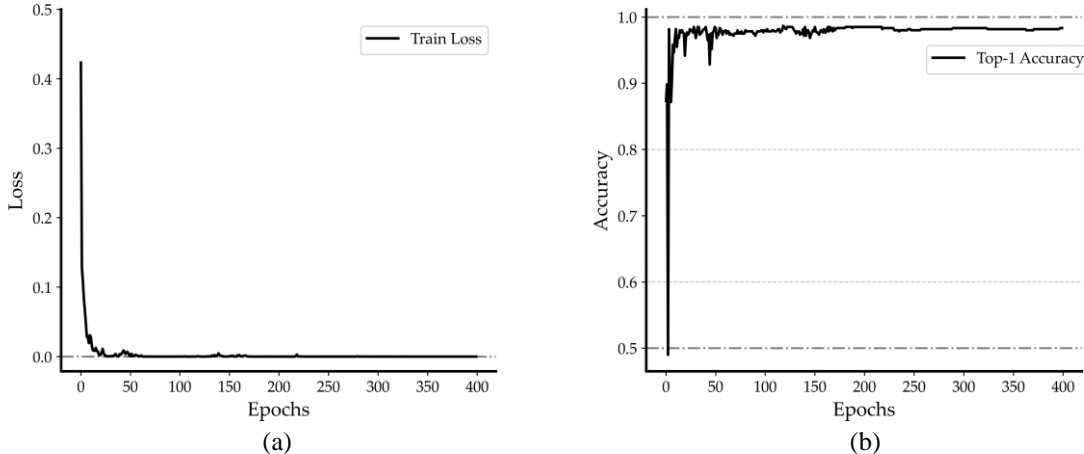

**Figure 5. YOLOv8 classification model training results. (a) Loss curve for network training; (b) Top-1 network training accuracy curve of the training set.**

After training completion on the self-built dataset, we tested the model's classification performance. These results cover four major cloud types, including Cirrus, Clear sky, Cumulus, and Stratus. Table 2 demonstrates the model's precision, recall, and F1 scores. As shown in the table, on the Self-built Dataset, the model delivers fairly steady classification performance for indistinctly bounded cumulus, maintaining relatively high precision, recall, and F1 scores of over 95%, indicating robustness and generalization capability of the model in categorizing cumulus. The model achieves outstanding classification efficacy on clear days, with all metrics reaching or approximating 100%, reflecting powerful generalization aptitude in recognizing clear conditions. The Cumulus type also sees all classification performance parameters surpassing 96%, denoting high classification accuracy. The Stratus category manifests extremely excellent outcomes on the Self-built Dataset across all metrics of 100%, implying that the model classifies Stratus very accurately with stable performance unaffected by dataset variations, successfully learning effective visual traits to discriminate the Stratus type.

When verifying our model's classification performance, we opted to validate on the public TCI Dataset to ensure extensive applicability of our model. Firstly, stringent quality control was imposed on the TCI dataset, removing images of inferior quality and ambiguous categorization. Eventually 900 high quality images per cloud type - Cirrus, Clear sky, Cumulus and Stratus - were screened, totaling 3600 images. Adopting identical training parameters as the self-built dataset, we trained the public dataset and validated performance on the test set containing 200 images per cloud type - Cirrus, Clear sky, Cumulus and Stratus - subsequently computing precision, recall and F1 scores for the model's classifications as depicted in Table 2. Evident from the table, the model demonstrates outstanding performance on the public TCI dataset, attaining commendable classification outcomes. Notably, for the Clear sky and Stratus types, the model approximates or achieves 100% accuracy across multiple evaluation metrics.

Compared to related research utilizing the bag of micro-structures (BoMS) approach for cloud type identification on the TCI dataset which encompassed five cloud types and attained an average accuracy of

93.80% after excluding mixed types (Li et al., 2016), our model realizes a higher average accuracy of 98.31% under the same assessment criteria. This further exhibits the superiority of our model architecture over preceding techniques, possessing more potent classification capability and performance. These results signify that our model framework not only manifests stellar performance on the self-built dataset, but can

also maintain lofty competency with robustness and generalization strengths on public data.

**Table 2. Performance comparison of cloud type classification and recognition: precision, recall and F1 score of public data sets and self-built data sets.**

| Cloud Type | Self-built dataset | | | Public cloud dataset | | | |
|---|---|---|---|---|---|---|---|
| | Precision (%) | Recall (%) | F1-Score (%) | Precision (%) | Recall (%) | F1-Score (%) | BoMS_Precision (%) |
| Cirrus | 95.45 | 98.00 | 96.71 | 94.71 | 98.50 | 96.57 | 87.20 |
| Clear sky | 100.00 | 98.67 | 99.33 | 100.00 | 98.50 | 99.24 | 99.50 |
| Cumulus | 97.30 | 96.00 | 96.65 | 98.52 | 100.00 | 99.25 | 92.00 |
| Stratus | 100.00 | 100.00 | 100.00 | 100.00 | 100.00 | 100.00 | 96.50 |
| Average | 98.19 | 98.17 | 98.17 | 98.31 | 99.25 | 98.77 | 93.80 |

On the four-type weather test set, five randomly selected images from each type were tested. As Figure 6

shows, all images obtained accurate category labels with confidence scores of 1.00, again validating the reliability of the training results in Table 2. Through training, the model has acquired the capability to discern different cloud morphologies based on visual characteristics like shape, boundary and thickness to generate cloud type classification outcomes. In summary, the model can not only effectively tackle various challenges in cloud classification tasks but also delivers consistent performance across cloud types on

validation and test sets. The robust overall performance provides a reliable cloud classification tool for practical applications.

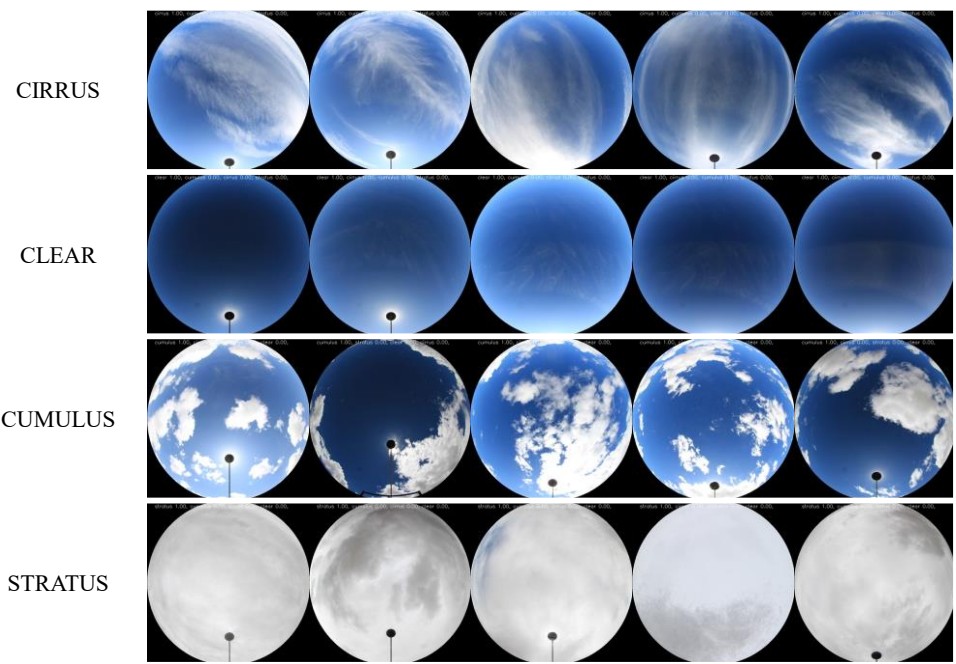

**Figure 6. Categorization effects of the four cloud dominant types.**

To further validate the model's discrimination of different cloud types, we computed the average
probability density functions of normalized red-blue ratio (NRBR) distributions for 1000 RGB images per
cloud dataset. Cloud image samples were then randomly drawn from each category and their NRBR
distributions derived. Finally, the Kullback–Leibler (KL) divergences between the sample distributions and
corresponding category averages were calculated. As Table 3 shows, the KL divergence between a sample
and its ground truth category is markedly lower than divergences to other categories. For instance, a clear
sky sample has an average KL divergence of 0.0357 to the clear sky category, but 11.2321 to the stratus
category. This signifies that the NRBR distribution of the clear sky sample identified by YOLOv8 aligns
closely with the true category average, with similar KL divergence relationships holding for other cloud
type samples. It verifies that the model can effectively discriminate the NRBR traits of different cloud types
to ultimately yield accurate cloud classification outcomes.

**Table 3. Verification of classification results by KL divergence. NRBR mean image: horizontal axis is Normalised RedBlue Ratio, vertical axis is probability density.**

| Isolated example / KL dispersion \ NRBR Mean | | Cirrus | Clear Sky | Cumulus | Stratus |
|---|---|---|---|---|---|
| Cirrus-997 | | 0.0623 | 2.1533 | 0.2079 | 5.2798 |
| Clear Sky-50 | | 1.1152 | 0.0357 | 1.0483 | 11.2321 |
| Cumulus-80 | | 0.6723 | 2.4738 | 0.0978 | 5.2600 |
| Stratus-148 | | 2.3976 | 7.4735 | 0.8972 | 0.0295 |

### 4.2. Cloud Recognition Effect

To improve the accuracy of subsequent cloud quantification, we first performed pre-processing enhancement on the whole sky images. However, considering different cloud types are impacted differently by illumination and haze, we designed an adaptive image enhancement strategy: applying lower intensity
for cirrus to preserve more edge details, while stronger intensity for other cloud types to eliminate overexposed areas. As shown in Figures 7a and 7b, this image enhancement algorithm makes the boundaries between clouds and blue sky more pronounced, with clearer ground objects and richer detail features.

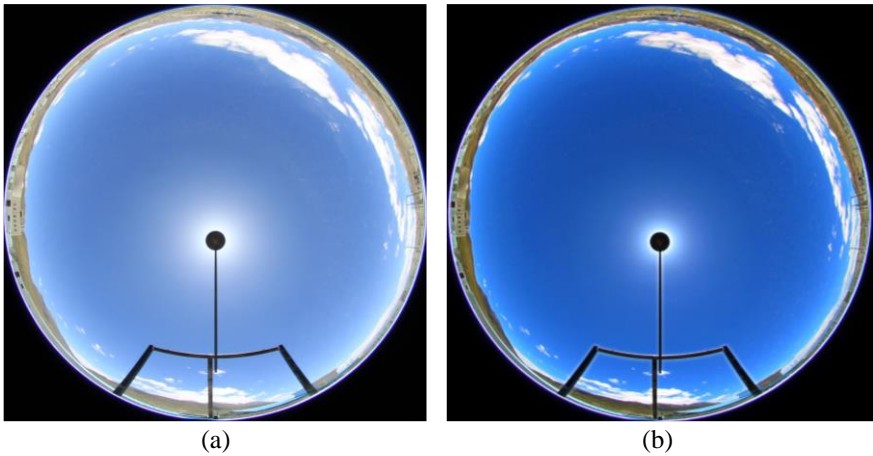

      (a)          (b)

**Figure 7. Comparison of image preprocessing effects. (a) Original image; (b) Image enhancement result.**

This study employs an adaptive finite sector segmentation strategy for feature extraction. For stratus and clear sky with distinct boundaries, just a few sectors are sufficient to accurately capture their traits. In

contrast, more sectors are utilized for the indistinct boundary cirrus and cumulus to enable more delicate partitioning that precisely seizes cloud edges. By further leveraging the k-means algorithm, we divide each sector region into three classes - blue sky, cloud and background. Compared to conventional holistic NRBR threshold segmentation segmentation, the segmentation tailored to cloud types has significantly better adaptivity and partitioning outcomes. As depicted in Figure 8, the finite sector segmentation and k-means

clustering achieve remarkable results in three challenging scenarios: (1) The bottom of thick cloud layers that are prone to misjudgment as blue sky by traditional methods; (2) The overexposed vicinity of the sun where RGB values resemble clouds, potentially causing some blue sky around the sun to be wrongly judged as white clouds by conventional techniques; (3) Thin edge areas of cloud layers that are difficult to accurately recognize by standard NRBR threshold segmentation, leading to deficient cloud quantification.

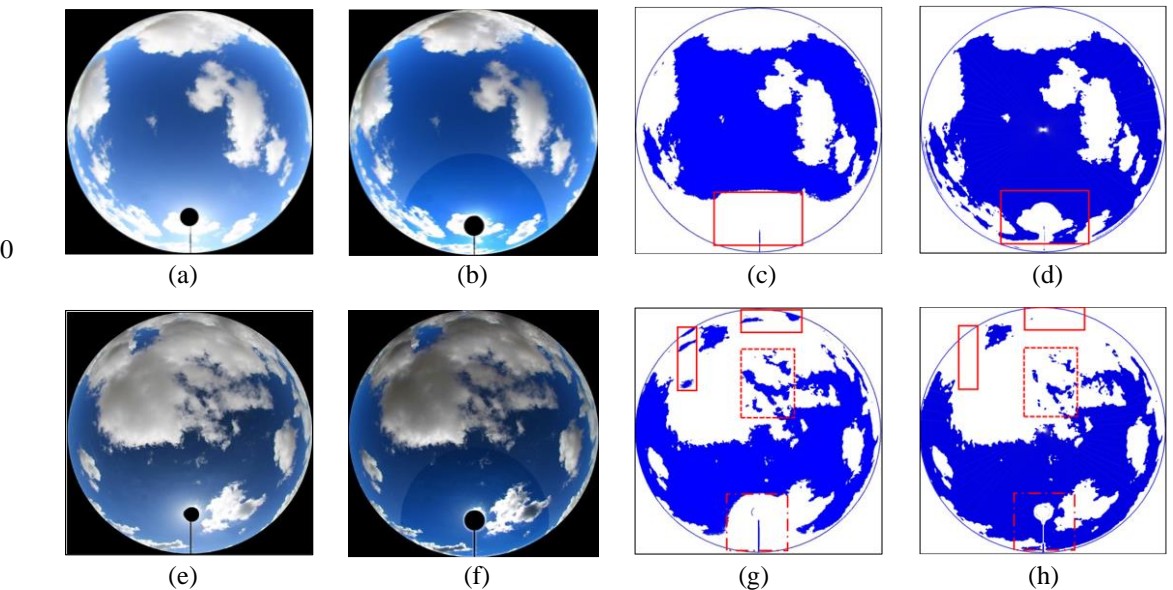

|  |  |  |  |
| :---: | :---: | :---: | :---: |
| (a) | (b) | (c) | (d) |
| (e) | (f) | (g) | (h) |

**Figure 8. Comparison of sector segmentation effects. (a/e) Cropped original image; (b/f) Adaptive enhancement**
**processing results; (c/g) Traditional NRBR threshold segmentation recognition processing results; (d/h) Finite sector segmentation k-means clustering results.**

    Through adaptive finite sector segmentation, we divide the original image into multiple sectoral regions, reducing the complexity of directly processing the entire image. This process enables the k-means clustering method to more effectively identify clouds in each sector, thereby significantly improving the

accuracy of detection. This forms the key strategy for our success in cloud amount calculation. As illustrated in Figure 9, the curve charts the cloud amount information at 15:00 every afternoon in June 2020 collected from valid images in Yangbajing area, comparing and analyzing the differences in cloud amount identification between the traditional NRBR threshold segmentation method and the image enhancement technique. Over the course of this month, after image enhancement, the cloud recognition effect has

conspicuous improvements for whole sky images with cloud cover below 80%. As denoted by the marked points in Figure 9, at 15:00 on June 17th, the cloud amount calculated from the enhanced cloud map has an approximated 40% higher precision than that obtained using the traditional NRBR threshold segmentation method. This is primarily attributed to NRBR threshold segmentation method solely relying on color

features, whereas the finite sector method synthesizes multiple features including shape and position for comprehensive judgment, hence possessing superior recognition effects on the overexposed areas surrounding the sun. Similarly, after image enhancement processing, as shown in Figures 8d and 8h, the cloud recognition effect at the bottom of thicker cloud layers and overexposed areas around the sun was significantly improved compared to Figures 8c and 8g.

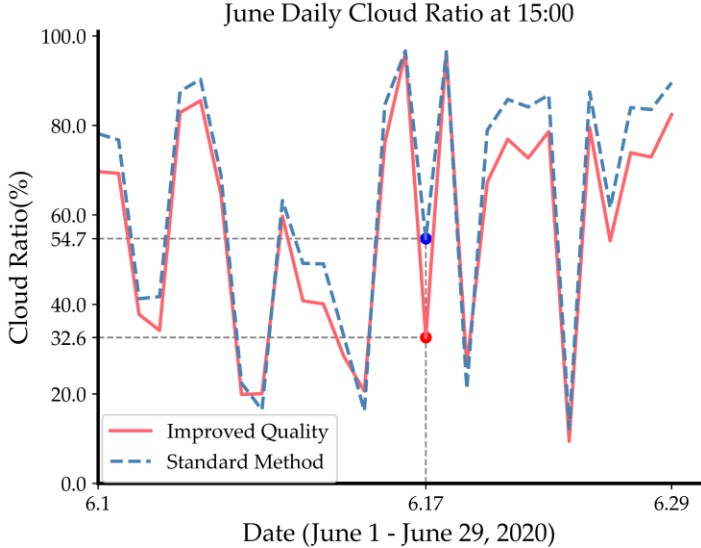

**Figure 9. Comparison of cloud cover between the traditional NRBR threshold segmentation method and the finite sector partitioned K-means clustering method in June 2020.**

**4.3. Spatial and Temporal Analysis of Cloud Types**

To gain deeper insight into the seasonal variations and diurnal patterns of cloud types in the Yangbajing area, Tibet, more detailed classification statistics on the 2020 annual daylight data are conducted. As depicted in Figure 10, stratus occurred most frequently throughout the year, accumulating 6622 times and accounting for 30% of total cloud occurrences. Clear sky and cumulus took the second and third places, appearing 5447 and 5365 times respectively, both comprising around 24%. Cirrus occurred least, at 5001 times making up 22%. This aligns with the climate characteristics of the Qinghai-Tibet Plateau. Stratus primarily form from atmospheric water vapor condensation, facilitated by the high altitude and greater atmospheric thickness in Tibet. Cirrus often develop at relatively lower altitudes and more humid climatic conditions (Monier et al., 2006), while the dry climate on the Tibetan plateau is less conducive to their formation. Analyzing by seasons, stratus appeared most in spring (March-May), occurring 2678 times and occupying 42.9% of all daytime cloud types in the season. Cumulus occurred most frequently in summer (June-August), reaching 3838 times and taking up 46.5% of total daytime cloud types. Clear sky dominated in autumn (September-November), appearing 2249 times and accounting for 42.8% of the daytime varieties. Winter (December-February) was also predominated by clear sky, which occurred 2150 times constituting 45.7% of the daytime population. The distinct seasonal shifts in cloud types across Tibet match its climate patterns - increased evaporation in spring facilitates thick cloud buildup; intense convection readily forms

cumulus in summer (Chen et al., 2012), aligning with the greater summer precipitation; while the relatively

dry, less snowy winters see more clear sky days. Analyzing diurnal fluctuations reveals that stratus and cumulus concentrate in afternoon hours, peaking at 17:00 and 18:00 for stratus (761 and 756 times respectively), and 13:00, 14:00 and 15:00 for cumulus (665, 707 and 684 times), potentially related to convective activity strengthened by afternoon surface heating. Clear sky occurrences are mainly distributed in the morning at 9:00, 10:00 and 11:00 (740, 812 and 743 times). Cirrus vary more evenly throughout the

510 day. Dividing the daylight hours of 7-20 into 7 periods, we statistically determine the peak timing of different cloud types: (1) Stratus peak at 17-18:00, occupying 39.1% of the total occurrences. (2) Cumulus peak at 13-14:00, taking up 34.2%. (3) Clear sky peaks at 9-10:00, constituting 40.6%. (4) Cirrus peak at 17-18:00, comprising 24.9%. The common afternoon emergence of cumulus may relate to intensified convective motions caused by daytime solar heating of the surface. Because the early air is less volatile and

515 has a lower water vapor content than other times of the day, clear skies are more common in the morning. Generally speaking, the development of cirrus requires relatively humid circumstances. Late afternoon solar radiation warms the surface and causes the air to rise, which aids in the vertical movement that carries water vapor to higher altitudes where it condenses as cirrus.

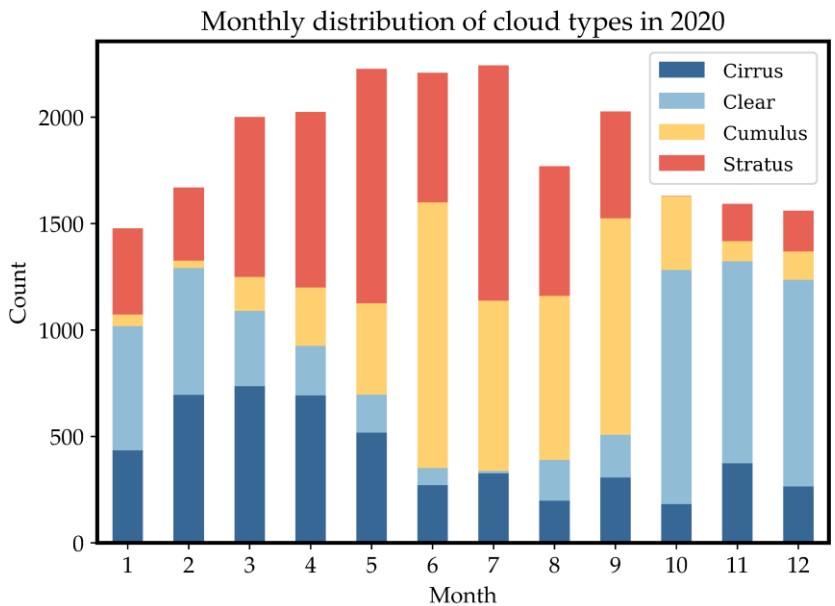

**Figure 10. Yangbajing Region's Monthly Distribution Trend of Four Types of Clouds for 2020.**

**5 Discussion**

**5.1 Comparison of Classification Performance**

Cloud detection and identification has long been a research focus and challenge in meteorology and remote sensing. Current mainstream ground-based cloud detection methods can be summarized into two categories – traditional image processing approaches and deep learning-based techniques (Hensel et al., 2021). Traditional methods like threshold segmentation and texture analysis rely on manually extracted features with weaker adaptability to atypical cases, whereas deep learning can automatically learn features for superior performance. This study belongs to the latter, utilizing the YOLOv8 model for cloud categorization to capitalize on deep learning's visual feature extraction strengths. Compared to other deep learning based cloud detection studies, the innovations of this research are three-fold: 1) An adaptive segmentation strategy tailored to different cloud types was designed, with segmentation parameters set according to cloud morphology to extract representative traits, improving partitioning accuracy. 2) Adaptive image enhancement algorithms were introduced, which markedly improved detection in regions with strong illumination impact like solar vicinity over conventional NRBR threshold segmentation. 3) Multi-level refinement was adopted to enhance capturing of cloud edges and bottoms. These aspects enhanced adaptivity to various cloud types under complex illumination. Limitations of this study include: 1) Small dataset scale containing only Yangbajing area samples due to geographic and instrumentation constraints; 2) Sophisticated model training and tuning demanding substantial computational resources; 3) Room for further improving adaptability to overexposed regions. Future work may address these deficiencies via enlarged samples, cloud computing resources, and more powerful models.

In the context of comparing YOLOv8 model against BoMS method on the TCI dataset for cloud type classification, while this study has indeed exhibited superior performance attributes, we acknowledge that novel image classification algorithms are consistently emerging. In recent scholarly work (Gyasi and Swarnalatha, 2023), a streamlined convolutional neural network (CNN) built upon MobileNet architecture has achieved substantial enhancements, reaching an overall accuracy as high as 97.45% on analogous public datasets. Similarly, other cloud classification networks such as CloudNet (Zhang et al., 2018), Transformer-based models (Li et al., 2022b), and Combined convolutional network (Zhu et al., 2022) have also demonstrated commendable classification efficacy. However, due to the lack of direct comparative empirical evaluations between these latest algorithms and our proposed YOLOv8 model within the current paper, it is not feasible to conduct a quantitative juxtaposition with these advancements. Despite this limitation, considering the cutting-edge achievements reported in the literature and the swift pace of technological progress within deep learning, future research endeavors will include meticulous comparative analyses of these state-of-the-art methods. This strategic move aims at rigorously validating and augmenting the robustness and generalization capabilities of our model under intricate meteorological circumstances, ensuring its continued competitiveness at the vanguard of research into cloud quantification. Ultimately, this drive is directed towards refining our existing framework continually and furnishing

climate science research with increasingly accurate and efficient solutions for cloud measurement tasks. Table 4 shows the comparison of our model with the most recent technical approaches in the literature.

**Table 4. Comparison of this study with the latest technological approaches in the literature.**

| Article | Dataset | Year | Model/Method | Accuracy(%) |
|---------|---------|------|--------------|-------------|
| Li et al. (2016) | TCI | 2016 | BoMS | 93.80 |
| Zhang et al. (2018) | CCSN | 2018 | CloudNet | 88.0 |
| Li et al. (2022) | ASGC | 2022 | Transformer | 94.2 |
| | CCSN | | | 92.7 |
| | GCD | | | 93.5 |
| Zhu et al. (2022) | MGCD | 2022 | Combined convolutional network | 90.0 |
| | NRELCD | | | 95.6 |
| Fabel et al. (2022) | All sky images (Owned) | 2022 | Self-supervised learning | 95.2 |
| Gyasi et al. (2023) | CCSN | 2023 | Cloud-MobiNet | 97.45 |
| Ours | All sky images | 2023 | YOLOv8 | 98.19 |
| | TCI | | | 98.31 |

560

**5.2 Model scalability**

Due to the limitations of single-site data in revealing the pattern of cloud cover change in a larger region, we decided to incorporate more data from meteorological stations with different geographic locations and climatic conditions in our future studies to enhance the model's generalizability to a wide range of geographic environments and climatic scenarios. We plan to build a dataset containing multi-site, cross-geographic cloud amount and cloud type data. By integrating and comparing data from different locations, we can not only validate and optimize the currently proposed cloud quantification method, but also assess its applicability and accuracy in different climatic contexts. The adaptive image enhancement strategy does not depend on specific lighting conditions and can be widely applied to various complex environments; the design idea of finite element segmentation combined with K-mean clustering can also be generalized to cloud computation in different regions, for example, inland regions where haze occurs more often can also be well applied. The modular design of this framework makes each component individually optimized and upgraded, which is very flexible.

In this study, although the example validation is only carried out at the Yangbajing station in Tibet, the method is highly scalable and universal, and the constructed end-to-end cloud recognition framework has the ability of generalization, and can be adapted to the cloud morphology characteristics of other geographic locations after appropriate model fine-tuning in the following ways:

(a) The climate characteristics of weather stations in different geographic locations are very different, such as high humidity in the tropics, extreme low temperature in the polar regions, and complex terrain in mountainous regions, for which the image preprocessing module needs to be adjusted as follows, (1) Climate-adapted image preprocessing: introduce region-specific light models and adjust the atmospheric light parameter A value in the image enhancement algorithm to adapt to the changes in the light under different climatic conditions, e.g., for the high latitude regions, the processing intensity of the defogging algorithm is strengthened to cope with the frequent fog and low-light conditions in winter; (2) terrain influence compensation: for mountainous or urban environments, the original zenith angle cropping range is modified to ensure that cloud identification is not interfered by surrounding environmental factors.

(b) Differences in all-sky camera models, resolutions and installation locations used by weather stations require the following adjustments to the reading module, (1) Modify the lens parameters in the algorithm configuration file, such as the image cropping range, the image suffix (e.g., jpg, png, etc.), and the image resolution standard. (2) Adjust the common data interface to ensure that the system can seamlessly access different brands and models of cloud cameras and data recording equipment to achieve automatic loading and standardized processing of data.

(c) Considering the specific needs of different weather stations, the system can provide highly personalized configuration options: (1) Parameter number configuration template: Provide preset parameter templates to set the optimal identification parameters and algorithm configurations for different climatic regions (e.g., tropical rainforests, deserts, and poles) and the frequency of occurrence of cloud types. (2) Dynamic adjustment mechanism: Dynamically adjust the algorithm parameters, such as the K value of K-Means clustering and the threshold value of cloud type identification, according to the system operation status and identification accuracy, in order to optimize the identification effect.

For overexposed regions: (1) plan to incorporate additional meteorological data, such as temperature, humidity, and wind speed, into our predictive models by combining these parameters with image data to refine our understanding of cloud formation dynamics and improve model accuracy under variable atmospheric conditions; (2) explore the temporal evolution of cloud patterns and their response to global warming trends, analyze historical and projected climate data to quantify how changes in temperature gradients, precipitation patterns, and atmospheric stability affect cloud morphology and distribution, and to develop models that can predict long-term changes in cloudiness, thereby contributing to climate prediction models; (3) To address the challenge of overexposure, we plan to investigate and implement state-of-the-art exposure correction algorithms, such as adaptive histogram equalization or high dynamic range (HDR) imaging, that can mitigate the effects of overexposure and thereby improve the accuracy of models under bright conditions. effects, thereby improving the model's ability to accurately identify cloud features under bright illumination conditions; (4) combining ground-based imagery with satellite data and potentially other remote sensing techniques can provide complementary perspectives on cloud cover and dynamics, and integrating these different data sources may enhance our ability to comprehensively model cloud systems, especially in regions where ground-based observations alone may not be sufficient.

## 5.3 Discussion on Clouds and Solar Radiation

An in-depth exploration of the relationship between cloud cover and solar radiation is a crucial aspect of our research. Different types of clouds, such as cirrus, cumulus, and stratus, have varying impacts on solar radiation. Generally, clouds absorb a portion of shortwave radiation, scatter another portion, and reflect the rest back into space, thereby altering the amount of solar radiation reaching the Earth's surface. Cirrus, due to their high altitude and composition of ice crystals, exhibit strong scattering of shortwave radiation and also significantly affect the emission and absorption of longwave radiation (Marsing et al., 2023; Shi and Liu, 2016). Cumulus, with their rough structure, contribute to strong scattering and some degree of absorption of shortwave radiation. Stratus typically form a thin and continuous layer, resulting in uniform attenuation of shortwave radiation. Rocha and Santos utilized machine learning techniques such as XGBoost and CNN-LSTM to process and analyze a large volume of image data provided by the GOES-16 satellite. They constructed a model capable of simulating global horizontal and direct vertical solar radiation intensity, capturing the complex effects of different cloud layers on the solar radiation field across various time and spatial scales. By learning and analyzing cloud features in satellite image data, researchers were able to more accurately estimate the impact of different cloud layers on solar radiation energy transfer at specific times and locations, thus revealing how clouds influence Earth's energy balance through radiation characteristics (Rocha and Santos, 2022). In a study by Matsunobu et al, CNN technology was employed to unveil unique visual features of cloud layers in remote sensing images. These features can effectively distinguish different cloud cover levels and classify the nature of cloud layers (Matsunobu et al., 2021). By identifying and quantifying the presence and distribution of clouds, it is possible to estimate the role of clouds in reflecting shortwave radiation and absorbing and re-emitting longwave radiation, contributing to an understanding of the role clouds play in the global climate system.

## 6 Conclusions

This research proposes a novel deep learning based whole sky image cloud detection solution, constructing a 4000-image multi-cloud dataset spanning cirrus, clear sky, cumulus and stratus categories that achieved markedly improved recognition and quantification outcomes in Tibet's Yangbajing area. Specifically, this study constructs an end-to-end cloud recognition framework. First, different cloud types are accurately determined using the YOLOv8 model with an average classification accuracy of more than 98%, and an average classification accuracy of more than 98% is also achieved on the TCI public dataset. On the basis of cloud classification, an adaptive segmentation strategy is designed for different cloud shapes, which significantly improves the segmentation accuracy, especially for convolutional clouds with fuzzy boundaries. Moreover, adaptive image enhancement algorithms were introduced to significantly improve detection in illumination-challenging areas around the sun. Finally, multi-level refinement modules based on finite sector techniques further upgraded judgment precision of cloud edges and details. Validation on the 2020 annual Yangbajing dataset proves stratus constitute the predominant type, appearing in 30% of

650 daytime cloud images, delivering valuable data support for regional climate studies. In conclusion, this framework significantly raises the automation level of ground-based cloud quantification to create a strong technological foundation for research on climate change. It does this by integrating various modules that cover classification, adaptive segmentation, and image enhancement. Additionally, it offers a referable paradigm for other cloud recognition tasks under complex lighting environments.

**Data availability**

The data that support the findings of this study are available from the corresponding author upon reasonable request.

**Author contribution**

YW and JL: conceptualization, methodology, developed the model code, formal analysis, investigation,
writing (original draft), writing (review and editing). YP and DS: conceptualization, resources, formal analysis, supervision. JZ, LW, WZ, XH, ZQ and DL: data curation, resources, methodology.

**Competing interests**

The authors declare that they have no conflict of interest.

**Acknowledgments**

This research was funded by the Second Tibetan Plateau Scientific Expedition and Research Program of China, grant number 2019QZKK0604; and by the National Natural Science Foundation of China (42293321 and 42030708). We would like to express our sincere gratitude to the Yangbajing General Atmospheric Observatory of the Institute of Atmospheric Physics, Chinese Academy of Sciences, and the
670 Chinese Academy of Meteorological Sciences for providing data for this study.

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
