# Peer review of "Innovative Cloud Quantification: Deep Learning Classification and Finite Sector Clustering for Ground-Based All Sky Imaging"

_EGUsphere, 2024_

## Author Comment (AC1)

**Response to Reviewer 1 Comments**

Thank you for your letter commenting on our manuscript entitled "*Innovative Cloud Quantification: Deep Learning Classification and Finite Sector Clustering for Ground-Based All Sky Imaging*" (MS No.: egusphere-2024-678). These comments are valuable and very helpful for the revision and improvement of our paper. We have carefully studied and made corrections, and hope to get your approval. The main changes of the paper and the responses to the review comments are as follows.

**Comments 1:** *By proposing this topic, the authors should know that the definition of clouds is challenging and observations of clouds from different instruments vary a lot, making cloud information uncertain. This brings a serious issue: how could the authors provide the true information for the training? Note that this question is general for all cloud identification studies.*

**Response 1:** Thank you very much for raising the issues of uncertainty in cloud definition and the potentially large differences in cloud information obtained by different observational instruments. In this study, we are fully aware of the difficulties posed by cloud definition and observation consistency, and have taken a number of measures to minimize the resulting uncertainties:

1. Data source and instrument calibration: The cloud observations we use come from ground-based all-sky imagers that have been rigorously calibrated to ensure the reliability and consistency of the underlying data. Meanwhile, we refer to the cloud classification standard of the International Meteorological Organization (WMO) to ensure that the definition of cloud types is accurate.

2. Multi-source data fusion and cross-validation: Although there may be errors in single instrument observations, we try to integrate data from different time periods and multiple observation platforms to reduce the bias caused by a single source through cross-validation, and strive to build a high-quality training set that contains a variety of typical cloud features.

3. Expert labeling and manual review: We invited meteorological experts to participate in the labeling process of cloud images to ensure that the training samples are accurately labeled. Meanwhile, the model prediction results were manually reviewed to further confirm the consistency between the cloud features learned by the model and the actual cloud patterns.

4. Adaptive and robust model design: To cope with changes in cloud morphology and lighting conditions, we developed adaptive segmentation and classification strategies and introduced image enhancement algorithms to ensure stable model performance in complex environments and minimize the impact of observational uncertainty on recognition results.

5. Validation and Comparison: Through comparison tests with the publicly

available dataset TCI, we confirm that the proposed method achieves a level higher than 98% in cloud classification accuracy, which indirectly verifies the validity and accuracy of the training data and the model we provided.

Thank you again for your valuable comments, which are extremely important guidance for our research.

**Comments 2:** *Regarding the importance of clouds, particularly on the radiation balance via its radiative forcing, a recent review study by Zhao et al. (2023, doi: 10.1016/j.atmosres.2023.106899) is worthy to mention here.*

**Response 2:** Thank you very much for your valuable comments and recommended literature, as you pointed out that the relevant studies on the importance of clouds in the radiative balance, especially through their radiative forcing, recently published in "Atmospheric Research" by Zhao et al. (2023, DOI: 10.1016/ j.atmosres.2023.106899) provides us with the latest research results and insights. Based on your suggestion, we have added the following to the Introduction's section on the background and significance of the study:

*"It is noteworthy that the critical role of clouds in the Earth's radiation balance has been further emphasized and empirically demonstrated in recent years. For example, Zhao et al. in their recent review explored in detail the impact of clouds on the global climate system through radiative forcing mechanisms, revealing how clouds act as a dynamic feedback system that can have a significant impact on the global radiation balance by playing a cooling role through blocking solar shortwave radiation as well as bringing a warming effect by absorbing and re-emitting longwave radiation (Zhao et al. 2023), this study reinforces the importance of quantitative cloud analysis for understanding and predicting climate change."*

Thank you again for your review and suggestions, which certainly enhanced the academic rigor and relevance of this paper.

**Comments 3:** *For sentence "In essence, clouds serve as an important "sunshade" to maintain the balance of the greenhouse effect and prevent overheating of the Earth": while the sentence is definitely correct, it is fair to mention the net cooling effect of clouds globally.*

Response 3: Thank you for your valuable suggestion that the issue of the net cooling effect of clouds should be mentioned, and we recognize that this point should be fully and accurately expressed. In order to improve the description of the article, we have revised the corresponding part of the original article as follows:

Original sentence, "Essentially, clouds act as a key 'sunshade' that maintains the balance of the greenhouse effect and prevents the Earth from overheating."

*Revised Sentence: "Clouds act as an important barrier in regulating the Earth's energy balance on a global scale, helping to prevent the Earth's surface from overheating, while also acting as a significant net cooling effect due to their nature of reflecting, absorbing, and emitting solar radiation, playing an integral role in the overall temperature regulation of the planet."*

In subsequent discussions, we will further elucidate this net cooling effect produced by clouds under different circumstances, as well as their complex interactions on the global climate and radiation balance, to ensure that readers gain a more comprehensive understanding.

**Comments 4:** *For sentence "For instance, high-level cirrus clouds mainly contribute to reflection and scattering, while low-level stratus and cumulus clouds more so cause the greenhouse effect": This is wrong, since high cirrus clouds play warming (greenhouse) effect and low clouds play cooling effect.*

Response 4: Thank you very much for catching the inaccuracies in my presentation of the effect of clouds on the Earth's radiation balance during your review. You are correct in pointing out that high altitude cirrus clouds actually exert a greenhouse effect, while lower stratus and cumulus clouds exert more of a cooling effect. Due to an oversight on my part that resulted in the description of these two cloud effects in the original article not matching the actual situation, we have corrected the corresponding sentence in the paper and the new formulation is as follows:

Original sentence: "For example, high-level cirrus clouds affect the radiative balance mainly through reflection and scattering effects, while low-level stratocumulus and cumulus clouds contribute more to the greenhouse effect."

*Revised Sentence: "For example, high altitude cirrus clouds actually contribute to the warming (greenhouse) effect on the Earth's radiation balance due to their stronger absorption and re-emission properties of longwave radiation, whereas low level stratus and cumulus clouds usually exhibit a cooling effect due to their good reflection and shading of solar shortwave radiation."*

Thank you again for your careful review and valuable comments, which play a vital role in improving the quality of the paper, and we will take this as an opportunity to more carefully check every scientific statement in the text to ensure the accuracy and completeness of the content. If you have any other comments or suggestions, please feel free to continue to put forward, so that we can further improve the paper.

**Comments 5:** *For sentence "Moreover, there are considerable regional disparities in cloud amount, and pronounced differences exist in regional climate characteristics": There are many studies regarding the regional variations of clouds which are worthy to refer here, such as a most recent study by Chi et al. (2024, doi: 10.1016/j.atmosres.2024.107316).*

Response 5: Your professional comments on the presentation of the paper regarding regional variability in cloud cover and its relationship with climate characteristics are sincerely appreciated. You pointed out that more studies on regional cloud amount variability should be cited to support this argument, especially the recent study by Chi et al. (2024) published in the journal (Atmospheric Research) (doi: 10.1016/j.atmosres.2024.107316). Based on your suggestion, we have revised the relevant sections and added the important research results of Chi et al. as references. Here is the revised sentence:

*Revised sentence: "There are large differences in cloudiness among different locations and significant differences in regional climatic characteristics, Globally, clouds over the oceans occur more frequently than over land, but the situation is reversed for cloud systems with more than two layers; seasonal variations in the global mean total cloud fraction are small but large among different latitudinal zones (Chi et al. 2024)."*

In addition, we will cite the Chi et al. study in detail at the appropriate places in the text, summarizing and discussing their findings in order to flesh out and strengthen the strength of the argument in this section.

**Comments 6:** *For image processing techniques used for cloud detection, previous studies should be introduced and cited, to identify the creativity of this study.*

Response 6: Thank you very much for your valuable suggestions. In the revised manuscript, we have fully recognized the importance of citing previous research to highlight the innovativeness of this study and have provided a detailed description and literature references of existing image processing techniques for ground cloud detection. The following is an overview of what we have added to the article:

*"In the field of meteorology and remote sensing, cloud detection and identification have been the core and challenge of research. The current mainstream ground-based cloud detection methods mainly include two categories: traditional image processing techniques and deep learning-based techniques (Hensel et al., 2021). Traditional threshold segmentation and texture analysis methods rely on manually extracted features that are less adaptable in dealing with atypical situations; while deep learning methods are able to automatically learn features for superior performance."*

Our research falls into the latter category and highlights the following innovations in particular: designing an adaptive segmentation strategy for different cloud types, which improves partitioning accuracy by extracting representative features by setting

the segmentation parameters according to the cloud morphology; introducing an adaptive image enhancement algorithm, which significantly improves the detection results, especially in the regions near the sun where the light influence is strong, and outperforms the traditional Normalized Differential Reflectance ( NRBR) segmentation method; the use of multilevel refinement technique improves the ability to capture the details of the edges and bottoms of various types of clouds, and enhances the adaptability to a wide range of cloud types under complex illumination conditions. We have not sufficiently discussed some specific image processing techniques used in previous studies and their limitations before, to compensate for this, in the subsequent revisions, we will compare and cite related studies in detail in order to further clarify the contribution of this study to the technological innovation of cloud detection. For example, the performance and limitations of methods such as the traditional threshold analysis method in specific scenarios will be described in detail, and the specific improvement measures and innovations of this study in terms of precise quantification of cloud amount and enhancement of classification accuracy will be clarified in comparison with the YOLOv8 model, adaptive segmentation strategy, and the cloud detection process combining the finite sector technique and k-means clustering adopted in this study.

Thank you again for your professional guidance, and we will incorporate the above additions when revising the paper to ensure that both the originality and innovativeness of this research work are reflected, and that research results in existing fields are fully respected and referenced.

**Comments 7:** *There are multiple previous cloud classification methods, including the machine learning algorithm, texture feature extraction, and so on, most recent studies should be mentioned or referred.*

Response 7: Thank you for your valuable suggestions on the scope of references to cloud classification methods in our study. In order to better highlight the innovation and rigor of this research, we have carefully reviewed and updated the descriptions and references to previous cloud classification methods in the text to reflect the latest research results and technological advances.

Based on the original text, we have highlighted the applications of machine learning algorithms in cloud classification in recent years. In particular, it is pointed out that Convolutional Neural Networks (CNNs) excel due to their ability to learn increasingly complex patterns and cloud texture properties from large-scale pre-training datasets, which addresses the shortcomings of traditional methods in characterizing and extracting cloud texture features. CNNs are able to capture the subtle textures of clouds, such as edges and shapes, using their hierarchical feature extraction framework, which leads to the effective classification of complex cloud patterning (Citation: Yu et al., 2020).

We also detail the wide application of unsupervised learning methods, especially k-means clustering, in cloud segmentation and recognition tasks. Several studies have utilized k-means models to rapidly cluster and identify clouds and clear-sky regions in all-weather imagery, significantly improving the speed and efficiency of cloud computation (Citation: Krauz et al., 2020). These unsupervised learning techniques simplify the workflow of cloud image analysis by autonomously discovering data category structure without manual annotation. In this study, we take full advantage of deep learning to realize the classification of four typical cloud types for the whole year of 2020 at the Yangbajing Observatory on the Tibetan Plateau with an accuracy of more than 95% through a customized version of the YOLOv8 architecture. Moreover, we innovatively designed a set of adaptive segmentation strategies for different cloud types, which significantly improved the performance of cloud classification and quantification under complex lighting environments by setting the segmentation parameters according to the cloud body morphology as well as eliminating the sunlight interference with an adaptive image enhancement algorithm.

*In light of your suggestion, we further enhance the citations of recent related studies, including but not limited to Zhang et al. (2018), Li et al. (2022b), Ma et al. (2021), Zhu et al. (2022), Gyasi and Swarnalatha. (2023), Li et al. ( 2017), He et al. (2018), Rumi et al. (2015), and Wu et al. (2021) on cloud classification, covering methods such as manual identification, threshold segmentation, texture feature extraction, and satellite remote sensing, etc., and analyzing in detail the strengths and limitations of the respective methods, so as to enable readers to better understand the role of the present study in solving the problem of climate scientific research in addressing the need for large number of fine cloud datasets with unique value and technological innovation.*

We will continue to monitor the latest progress in this field and add and update the corresponding literature citations in the final manuscript to ensure that the current state of the art in cloud classification technology and the unique contributions of this study are fully presented.

**Comments 8:** *Laser radar does not necessarily have large equipment size.*

Response 8: With regard to your reference to the fact that LIDAR does not necessarily have the dimensions of a large-scale device, we fully share your viewpoint. In the original presentation, the general characteristics of LIDAR systems may have been described in too general a manner, ignoring the development trend and technological progress of individual miniaturized or portable LIDARs. Therefore, we will clarify and correct the dimensions and forms of LIDAR in the corresponding section to ensure the accuracy of the presentation.

Original: "Laser radar emits sequenced laser pulses and estimates cloud vertical structure and optical depth from the backscatter to directly quantify cloud amount, but

has large equipment size, high costs, limited coverage area, and cannot produce cloud distribution maps."

*Revised sentence: "Lidar can directly quantify cloud amount by emitting sequential pulsed lasers and estimating cloud vertical structure and optical thickness from the backscatter information. While miniaturized or even portable Lidar equipment exists in the market, these instruments have high costs and limited coverage area in the all-sky cloud image recognition method involved in this study."*

Thank you again for your efforts to improve the quality of the paper, and we will make sure that the amended text reflects more objectively and fairly the characteristics and development of LiDAR technology.

**Comments 9:** *"with relatively good air quality and low atmospheric pollution levels": I think using "with relatively good air quality" is enough.*

Response 9: Dear reviewer, regarding the descriptive problem you pointed out, you think that the expression "relatively good air quality" is sufficient to express "the region has relatively good air quality", and there is no need to mention the additional phrase "low air pollution levels". There is no need to refer to "low levels of air pollution". We agree that this is a more concise and clearer expression that can directly convey the key message. Therefore, we have revised the original draft as follows:

*Revised sentence: "The Yangbajing area is far away from industries and cities, and the air quality is relatively good, which can reduce the impact of atmospheric pollution on cloud observation"*

Thank you again for your careful guidance, which has helped to improve the quality of our paper.

**Comments 10:** *Table 1: "Measure cloud distance" is better as "Measurable cloud distance"*

Response 10: Thank you very much for your careful review of the table in the manuscript and your valuable suggestions. In response to your suggestion, we fully accept it and change the title of the column "Measure cloud distance" to "Measurable cloud distance" to more accurately reflect the actual meaning of the indicator, i.e., the maximum distance range of cloud cover that can be measured by the equipment. The column heading of "Measure cloud distance" will be changed to "Measurable cloud distance" to more accurately reflect the actual meaning of the indicator, i.e., the maximum range of cloud distance that can be measured by the device. The revised table is shown below:

| Function | Description |
| --- | --- |
| Measurable cloud distance | 0~10Km |
| Measuring range | Elevation angle above 15° |
| Observation periods | Observe every 10 minutes |
| Horizontal visibility | ≥2km |
| Operating temperature | -40°~50° |
| Sensor | CMOS |
| Image resolution | 4288 × 2848 |
| Operational durability | 24 h operation |
| Ingress protection | IP65 |

We value your review comments and have revised the manuscript accordingly, as appropriate, to enhance the rigor and accuracy of its presentation.

**Comments 11:** *3.2.3: Have similar indicators been used by other studies? If have, a few reference could be helpful.*

Response 11: Thank you very much for your valuable comments on the use of evaluation metrics in Section 3.2.3. Comparison and citation of similar evaluation metrics used in similar studies is essential to validate the reliability and validity of the methodology of this study. During the cloud classification performance evaluation process, we did adopt the industry widely recognized metrics of Precision, Recall and F1 score, which have been used in several previous studies to measure the performance of cloud classification systems, e.g., studies such as (Dev et al., 2017; Guo et al., 2024) have used similar evaluation system. In the revised manuscript, we will explicitly point this out and cite relevant literature to support the rationality of our choice of these metrics, demonstrating the consistency and comparability of this study with existing work.

The following are examples of some of the reference citations that are planned to be included:

*1. Dev, S., Lee, Y. H., and Winkler, S.: Color-Based Segmentation of Sky/Cloud Images From Ground-Based Cameras, IEEE J. Sel. Top. Appl. Earth Obs. Remote Sens., 10, 231-242, 10.1109/JSTARS.2016.2558474, 2017. Precision, recall were also used as one of the criteria for evaluating the performance of cloud segmentation algorithms in that study.*

*2. Guo, B., Zhang, F., Li, W., and Zhao, Z.: Cloud Classification by Machine Learning for Geostationary Radiation Imager, IEEE Trans. Geosci. Remote Sens. . , 62, 1-14, 10.1109/tgrs.2024.3353373, 2024. where metrics of precision, recall, and F1 score are used to evaluate cloud classification models.*

Through literature citation and illustration, we believe that we can better demonstrate that the evaluation metrics chosen in this study are consistent with peer studies and facilitate readers in understanding and evaluating the results of this study in the cloud classification task.

**Comments 12:** *3.4: As indicated, a proper K value is important for K-means method. How do the authors choose their K values?*

Response 12: Thank you very much for your attention and guidance in submitting the paper on the issue of K-value selection in K-mean clustering methods. In Section 3.4, we indeed did not fully elaborate the process of determining the K-value, for which we apologize and will add and improve it in the revised manuscript with the following modifications:

*After obtaining the cloud type adaptive segmented images, for the K-mean clustering within each sector, we executed several trials to determine the optimal K-value. The specific selection process is as follows:*

*(1) Initial estimation: a preliminary K-value setting is performed based on the complexity of the observed data and the expected number of clustering categories (e.g., sky, cloud, and background).*

*(2) Iterative optimization: By implementing the K-mean algorithm and observing the clustering results, the K-value is adjusted according to the actual clustering effect until the clustering results are stable, i.e., the clustering centers are no longer significantly changed between several adjacent iterations (Dinc et al., 2022).*

*(3) Evaluation indexes: using clustering effectiveness indexes such as contour coefficient, Calinski-Harabasz index, Davis-Boulding index, etc., the clustering results under different K-values are evaluated, and the K-values that make the evaluation indexes optimal are selected.*

*(4) Evaluation index: Combining the knowledge and practical experience of meteorological experts, the selected K-values are tested for their rationality to ensure that they are in line with the principles of meteorology and actual observation.*

*In this study, for the task of cloud quantification and classification of all-sky images in the Yangbajing area, we chose k=5 as the hyperparameter of the clustering algorithm, which is based on a series of rigorous experimental analyses and the conclusion of practical effect evaluation. Through the trial and error and cross-validation of a large number of sample data, we found that when k is set to 5, the clustering results can most effectively distinguish the blue sky, the white cloud layer, the transition zone and possible ground or near-ground occlusions, thus achieving the desired segmentation effect. We also draw on the a priori knowledge in the field about the identification of cloud amount and cloud features, and combine it with the field observation data to ensure that the selected k value matches the actual physical phenomena. The clustering strategy is able to maintain a high level of robustness and identification effectiveness under a variety of lighting dynamics in the Yangbajing area, where the lighting conditions are complex and changeable.*

In the revised manuscript, we will document and clearly articulate this selection process for readers and peer reviewers.

---

## Author Comment (AC2)

**Response to Reviewer 2 Comments**

Thank you for your letter commenting on our manuscript entitled *"**Innovative Cloud Quantification: Deep Learning Classification and Finite Sector Clustering for Ground-Based All Sky Imaging**"* (MS No.: egusphere-2024-678). These comments are valuable and very helpful for the revision and improvement of our paper. We have carefully studied and made corrections, and hope to get your approval. The main changes of the paper and the responses to the review comments are as follows.

**Comments 1:** *The review of existing traditional cloud detection methods is not comprehensive enough, and a more systematic evaluation of their strengths and limitations is needed.*

**Response 1:** Thank you very much for your valuable suggestions on the review section of traditional cloud detection methods. In the original manuscript, we indeed did not provide a comprehensive and systematic assessment of the existing traditional cloud detection methods, especially failing to fully elaborate on the advantages and limitations of each method. To improve this, we will deepen and expand this section in the revised manuscript to ensure that readers can gain a more complete and in-depth understanding.

*In the "Introduction" chapter of the new revised version, we will introduce the application of traditional image processing techniques in cloud detection, including threshold segmentation and texture analysis, compare their performance in different scenarios, and analyze their adaptations in dealing with complex lighting conditions, cloud diversity, and ground object occlusion, etc., and analyze their adaptability and limitations when dealing with complex lighting conditions, cloud type diversity, and ground occlusion. At the same time, we will cite more related research literature to reflect the comprehensiveness and objectivity of the evaluation of existing methods.*

*In addition, we will especially emphasize the characteristics of traditional methods in data processing and real-time, such as the advantages of cloud radar in vertical structure detection and satellite remote sensing in large-area coverage, while pointing out their deficiencies in resolution, local small-scale cloud detection, and light sensitivity. Through such comparative analysis, we will be able to highlight more clearly the innovation and necessity of adopting deep learning and adaptive segmentation strategies in this research.*

Once again, we thank you for your professional guidance in this research direction, and we will fully incorporate your comments in the upcoming revisions, with a view to making this paper a stronger demonstration of the advancement and practicability of our proposed method based on evaluating and comparing traditional cloud detection methods.

**Comments 2:** *Although data from the Tibetan Plateau site was used, the spatial representativeness is still limited due to the use of a single site. Future work should consider incorporating data from multiple regions to enhance the model's broad applicability.*

**Response 2:** Dear reviewer, you have pointed out that this study only uses data from the Yangbajing station on the Tibetan Plateau, so there are some limitations in spatial representativeness, which is a direction we should focus on when further improving and expanding our research work in the future. We also recognize the limitations of single-site data in revealing the pattern of cloud cover change over a larger region, and we have decided to incorporate more data from meteorological stations with different geographic locations and climatic conditions into our future studies to enhance the generalizability of the model to a wide range of geographic environments and climatic scenarios. *We plan to build a dataset containing multi-site, cross-geographic cloud amount and cloud type data. By integrating and comparing data from different locations, we can not only validate and optimize the currently proposed cloud quantification method, but also assess its applicability and accuracy in different climatic contexts.*

In the next iteration of the study, we intend to actively collaborate with other meteorological observatories to share and integrate all-sky imaging data from multiple meteorological stations around the globe, with the aim of creating a large dataset that is more reflective of the diversity of global climatic features and cloud variability. The purpose of doing so is to further enhance the value and credibility of the application of automatic cloud identification and quantification techniques in global climate research.

We thank you again for your review and guidance, and we have made substantial improvements in the "Discussion" section of the revised manuscript in response to this suggestion, and will fully reflect these improvements in future revisions of the paper.

**Comments 3:** *While the association between cloud amount and solar radiation is mentioned, no in-depth discussion is provided. It is recommended to further analyze the influence of different cloud types on solar radiation characteristics.*

Response 3: Your valuable suggestions on this study are sincerely appreciated. You have pointed out that although the correlation between cloudiness and solar radiation is mentioned in the paper, the extent to which the relationship between the two is explored in depth is not yet sufficient, in particular the lack of a specific analysis of the effect of different types of cloud cover on the solar radiation characteristics. We fully recognize your comments and set out for you in this response how we plan to improve this section.

*In the Discussion section of the revised manuscript, we plan to discuss the effects of different types of clouds on solar radiation characteristics, analyzing in detail how cirrus, cumulus, stratocumulus, and clear-sky conditions can alter the Earth's energy balance through their different absorption, scattering, and reflection characteristics of shortwave and longwave radiation. We will analyze a large amount of satellite image data to construct a solar radiation model using machine learning techniques such as XGBoost, CNN-LSTM, etc., in conjunction with the research results of Rocha and Santos (2022), in order to deeply investigate the*

*mechanism of the various types of clouds affecting the solar radiation in the temporal and spatial dimensions. Meanwhile, the cloud detection technique proposed in this study is utilized to more precisely quantify the blocking and greenhouse effects of different types of clouds on solar radiation flux, especially how different types of clouds affect surface temperature and energy balance in different seasonal and regional contexts. In addition, we will draw on the research ideas of Matsunobu et al. (2021) to visualize the specific effects of different cloud amounts on the solar radiation balance by analyzing the unique visual characteristics of cloud cover in remote sensing images.*

We will practically implement your suggestions in the subsequent revisions to enhance the overall academic value and impact of the paper.

**Comments 4:** *Although the methodology is clearly presented, some details regarding equations, parameters, and symbols are not comprehensively explained, requiring further elaboration and clarification to ensure the reproducibility and transparency of the research work. Specifically: An explanation of the variables TP, FP, TN, and FN used in the evaluation metrics such as precision, recall, and F1-score, along with their calculation methods, should be provided to facilitate better understanding of the evaluation system.*

Response 4: Thank you very much for your valuable suggestions on the calculation methods of cloud classification performance evaluation metrics in this paper. Based on your review comments, we will further clearly explain the three core variables used in the evaluation metrics, TP, FP, and FN, as well as their specific calculation methods in the revised paper to enhance readers' understanding of the whole evaluation system.

In the context of cloud classification tasks, we define the following:

*True Positive (TP): The actual number of positive samples correctly predicted by the model (i.e., cloud category), representing the number of true cloud images identified by the model.*

*False Positive (FP): The number of samples incorrectly predicted as positive but actually belonging to the negative class (non-cloud category), indicating the number of cloud images erroneously classified by the model.*

*False Negative (FN): The number of samples incorrectly predicted as negative but actually belonging to the positive class, representing the number of cloud images missed by the model.*

We have explained these concepts and their calculations in detail in the "Cloud Classification Evaluation Indicators" section of the revised draft so that readers can better grasp the performance evaluation criteria of the model in cloud classification tasks.

**Comments 5:** *Although the methodology is clearly presented, some details regarding equations, parameters, and symbols are not comprehensively explained, requiring further elaboration and clarification to ensure the reproducibility and transparency of the research work. Specifically: The details of the image enhancement algorithm for dehazing need to be thoroughly described, especially the processes for obtaining the key parameters, atmospheric light A and transmission rate t, to ensure the reproducibility of the image enhancement step.*

Response 5: Thank you for your attention and valuable suggestions on the details of image enhancement algorithm de-fogging in this paper. In the revised paper, we have fully responded to your request by describing in detail the key steps of the de-fogging process and the parameter acquisition method to ensure the reproducibility of the image enhancement session. Our image enhancement algorithm adopts the dark channel prior algorithm, and its main process is as follows:

*(a) Computing the dark channel image: For each pixel in the input image, the dark channel image is computed by selecting the minimum value among its RGB channels. The dark channel image reflects the minimum brightness within pixel regions, where low brightness regions typically correspond to areas containing haze, providing us with clues for estimating haze information.*

*(b) Estimating the global atmospheric light A: The global atmospheric light intensity A is estimated using the minimum non-zero value in the dark channel image. Atmospheric light serves as the background light source that affects the overall scene brightness, playing a crucial role in the haze scattering model.*

*(c) Obtaining the transmission rate t: Based on the atmospheric scattering model, the transmission rate t is calculated for each pixel in the image, representing the visibility of the pixel. The transmission rate reflects the extent of haze's impact on light propagation.*

*(d) Applying the dehazing formula: The dehazed enhanced image J(x) = I(x) * (1 - A)t + A is applied, where J represents the dehazed image, and I is the original input image. Through this dehazing algorithm, haze in the image can be effectively removed, making cloud layers and the boundary of the blue sky more distinct, which is beneficial for generating high-quality cloud cover data.*

*In particular, when dealing with different types of cloud layers, we have devised adaptive enhancement strategies for varying cloud thicknesses. For instance, for thin stratocumulus clouds, to avoid excessive enhancement and filter out cloud layer details, a smaller atmospheric light value A is chosen. In contrast, for thicker cumulus, stratocumulus, and clear sky images, a larger atmospheric light value A is used to enhance the removal of overexposed areas and achieve a more uniform sky distribution.*

Thank you again for your review and guidance, and we believe that the requirement of ensuring the reproducibility of the image enhancement steps has been fulfilled by the above detailed description. If necessary, we can also provide more detailed algorithm implementation steps and parameter adjustment basis for peer scholars' reference and verification.

**Comments 6:** *Although the methodology is clearly presented, some details regarding equations, parameters, and symbols are not comprehensively explained, requiring further elaboration and clarification to ensure the reproducibility and transparency of the research work. Specifically: The finite sector K-means clustering segmentation strategy employs different numbers of sectors for different cloud types, but the rationale and basis for this setting are not explained. The authors should clarify the reasons behind the chosen sector numbers for each cloud type.*

Response 6: Thank you for your valuable comments on the paper, particularly regarding the rationale and basis for setting different numbers of sectors for different cloud types in the finite sector K-means clustering segmentation strategy. Based on your feedback, we recognize the need for a more detailed explanation of this key design decision. In the study in this paper, we have designed different sectorization schemes for each of the four typical cloud patterns - cirrus, clear sky, cumulus, and stratocumulus. This differentiated setup is based on the following rationale:

*(a) Cirrus clouds, due to their weak shape and color similarity to the sky, pose significant identification challenges. To capture cirrus cloud features more finely, we segment the entire sky image into 72 sector areas. More sectors aid in extracting subtler color and texture variations, thereby enhancing the clustering algorithm's accuracy in distinguishing cirrus clouds from other celestial elements.*

*(b) Clear sky images, containing fewer elements, require only 4 sectors for effective differentiation. This avoids unnecessary subdivisions, reducing computational complexity and enhancing algorithmic execution efficiency and classification accuracy in simple scenes.*

*(c) Cumulus clouds exhibit distinct edges, but uneven lighting may cause visual disturbances. To balance edge information capture and internal structure consistency, we divide them into 36 sector areas. This ensures both cloud boundary recognition and adaptation to potential lighting differences within cumulus clouds.*

*(d) Stratocumulus images consist of relatively few and evenly distributed elements. Therefore, they are also divided into 4 sectors to meet the clustering analysis requirements, maintaining necessary spatial resolution while avoiding noise and redundant calculations resulting from excessive sectorization.*

*The selection of these sectors is based on a large amount of measured data and an in-depth understanding of cloud morphology, and we experimentally verified that these adaptive segmentation strategies significantly improve the accuracy of the clustering algorithm in identifying different types of cloud cover. In the "3.4 Finite Sector Segmentation and K-means Clustering" section of the revised paper, we will further clarify the theoretical basis for choosing a specific number of sectors for each cloud type, in order to let the readers understand and agree with our methodological foundation more comprehensively. We hope that readers will more fully understand and agree with the basis of our methodology.* Thank you again for your review and suggestions, and we look forward to answering your questions and improving the quality and scientific value of the paper in the revised manuscript.

---

## Author Comment (AC3)

**Response to Reviewer 3 Comments**

Thank you for your letter commenting on our manuscript entitled *"Innovative Cloud Quantification: Deep Learning Classification and Finite Sector Clustering for Ground-Based All Sky Imaging"* (MS No.: egusphere-2024-678). These comments are valuable and very helpful for the revision and improvement of our paper. We have carefully studied and made corrections, and hope to get your approval. The main changes of the paper and the responses to the review comments are as follows.

**Comments 1:** *In the title, abstract and introduction of the manuscript, "cloud quantification" has appeared many times, but without a clear definition.*

**Response 1:** Dear reviewers, Thank you for your valuable comments on our manuscript, in which you pointed out that we have mentioned the term "cloud quantification" several times in the title, abstract, and introduction without defining it clearly. We appreciate your careful review and constructive feedback. We will add a clear definition of "cloud quantification" in the introduction of the revised manuscript to ensure that readers can accurately understand the importance of the concept and its role in climate research. The changes are summarized below:

*"Cloud quantification is the precise analysis of sky images to transform cloud body characteristics into a series of quantifiable parameters, including but not limited to cloud amount and cloud type, which are essential for understanding and modeling the Earth's radiation balance, energy transport, and climate change."*

We hope that the revised manuscript will better introduce the concept of "cloud quantification" to our readers, and thank you again for your careful review.

**Comments 2:** *In the second paragraph of the introduction, the structure read somewhat chaotic. The classification method and observation instruments are also mentioned in the overview of cloud classification method. In the later description of the cloud quantification method, the advantages and disadvantages of the existing cloud quantification methods are not specifically introduced. It is suggested to reconsider the structure of this part.*

**Response 2:** Dear reviewers, we agree with your comments about the slightly confusing structure of the second paragraph of the introduction, the mixing of classification methods and observation instruments in the overview of cloud classification methods, and the failure to introduce the strengths and weaknesses of existing cloud quantification methods in the subsequent description of cloud quantification methods, and will conduct a comprehensive reorganization and optimization of the content of this

part of the paper.

In the revised version, we will set up a separate subsection to systematically analyze the strengths and weaknesses of various cloud quantification methods, and adjust the overall structure of the introduction section to ensure that it starts from the climatic impacts of cloud phenomena to the scientific significance of cloud classification and quantification, and gradually transitions to the problems of the current technological tools and the methods proposed in this study to address these problems. It is hoped that the revision will enhance the coherence and hierarchy of the exposition, and allow readers to grasp the background and innovation of the study more clearly. The additions are as follows:

*"The advantages of traditional image processing techniques are mainly reflected in the easy operation and low computational cost, which are suitable for rapid preliminary identification of cloud cover areas, however, the high sensitivity of such methods to changes in lighting conditions leads to unstable identification results under complex lighting dynamics, especially in the identification of high-altitude thin cirrus clouds, complex boundary cloud bodies, and multiple clouds, due to the lack of adaptive ability and accurate feature expression, it is difficult to achieve the ideal quantization accuracy and weak adaptability to atypical cloud types, which affects the accuracy of cloud calculation. Deep learning methods can efficiently and accurately classify and segment cloud images under complex cloud types and various lighting conditions by means of a deep neural network model driven by large-scale training data, and significantly improve the quantization performance under unfavorable lighting environments by combining with algorithms such as image enhancement. Deep learning methods also have obvious shortcomings, such as relying on a large amount of labeled data, high-performance computational resources, and the recognition performance in extreme lighting scenarios such as extremely bright or dark still needs to be improved."*

**Comments 3:** *In the abstract, the traditional NRBR recognition method may be just summarized in the introduction. In the last paragraph of the introduction, it mentioned the problem of cloud identification in current algorithm, but it is not clear which specific method the author referred to?*

**Response 3:** Thank you for your valuable comments, and based on your suggestion, we have, in the abstract section, in order to keep the content focused and compact, I have moved the specific description of the traditional NRBR recognition approach to the appropriate place in the introduction section so that the reader can have an overview understanding of this basic recognition approach before entering the main text. The limitations of the current algorithms for the cloud recognition problem, mentioned at the end of the introduction, have now been clearly referred to and detailed. The last paragraph of the original introduction has been modified as follows:

*"Currently, many cloud recognition algorithms face significant challenges in dealing with different cloud types, especially high-altitude thin cirrus and transitional hybrid clouds (Ma et al., 2021). Among them, the traditional NRBR (Normalized Red/Blue Ratio) identification method, although able to provide preliminary cloud estimation in general, shows obvious*

*limitations in terms of shadowing effects and identification of thin cirrus edges due to the fact that it relies only on color features to make judgments, and the variation of illumination conditions greatly affects the identification results."*

We will also further detail the recognition limitations of the existing methods in different scenarios in the subsequent chapters to ensure that the thesis is clearly and accurately presented. Thank you again for your help and support in improving the quality of the thesis. If you have any other suggestions or questions, please feel free to continue your guidance.

**Comments 4:** *In Section 2.3, how are the four types of data divided in image dataset?*

**Response 4:** Dear reviewers, thank you for your valuable comments on the paper. Regarding your question in Section 2.3 about how the four categories of data in the image dataset are categorized, I have made the following changes to the description in the original paper:

*"We started by dividing the 4000 rain- and snow-free, unobstructed, high-quality all-sky images into four categories of 1000 images each, which are: cirrus, sunny, cumulus, and stratocumulus clouds; it should be emphasized that the division of clouds into the four main types is done here in order to accurately quantify the proportion of clouds in each category, rather than considering mixed clouds. These four cloud types play an important role in the region's weather and are the main references for this categorization."*

If you have any other suggestions, please feel free to let us know and we will make more detailed notes or changes accordingly. Thank you again for your review and guidance!

**Comments 5:** *In Section 3.2.1, the description of neural network design is not clear. What are the advantages of the YOLOv8 architecture in solving the research problems of the current manuscript, and why choose the framework? In addition, YOLOv8 involves the convolution part, in which the process has certain requirements on the size of the input image. Why is the input image size of 680×680 selected in this paper?*

**Response 5:** Thank you for your interest in the neural network design section of subsection 3.2.1 of the article and for your valuable suggestions. We have enhanced the description of the advantages of the YOLOv8 architecture in solving the current research problem, and the following modifications and responses are made to address the issues you raised:

*"The main reason why YOLOv8 is the preferred framework in this study is its unique design that can effectively handle the task of all-sky image cloud classification under complex lighting conditions. Compared to the previous YOLO series and some other classical image recognition models, YOLOv8 is able to extract richer gradient flow information by adopting Darknet-53 as the Backbone and utilizing the modified C2f module to replace the original C3 module (Li et al.,*

*2023), which is conducive to capturing the cloud's delicate textural and boundary features. Meanwhile, the PAN-FPN structure of YOLOv8 achieves model lightweighting while retaining the original high-performance performance, while the detection head part adopts a decoupled structure, which is responsible for the classification and regression tasks, respectively (Xiao et al., 2023), and adopts the binary cross-entropy loss (BCE Loss) for the optimization of the classification task, together with the distributed focus loss (DFL) and the complete IoU loss (CIoU) for bounding box regression prediction, this detection structure can significantly improve the detection accuracy and convergence speed of the model (Wang et al., 2023)."*

Regarding the selection of the input image size, we set it to 680 × 680 pixels, which is because the convolutional part of the YOLOv8 network does have some requirements on the input image size. This size was selected based on the consideration of several factors:

(1) The original all-sky image has extraneous black background as well as feature interference, and the resolution of 680 × 680 is the result of compressing the image after removing the image edges in a specified range of zenith angles.

(2) The resolution of 680 × 680 retains the main detailed features of the clouds in the image while significantly reducing the file size, which is beneficial to the loading and computational efficiency of the model in the training phase;

(3) This size not only meets the demand of YOLOv8 network structure on the image input size, but also takes into account the various morphological features of the cloud body in the image, which ensures that the model is able to maintain good recognition performance when dealing with cloud bodies at different scales and under complex lighting environments.

Thank you again for your review and questions, and we hope that these improvements will shed more light on the applicability of the YOLOv8 architecture in this study and the reasonableness of the selected input image sizes. Please feel free to give us feedback if there are any other areas that need further clarification or refinement.

**Comments 6:** *For what reason is the training epoch set to 400? Because in Figure 4, when the epoch is greater than 200, it is found that F1 is basically unchanged, and the loss is no longer reduced.*

**Response 6:** Thank you for your valuable comments on the experimental parameter setting section. Regarding the reason for setting the number of training rounds to 400, we did observe in the initial experimental phase that the training process stabilized the F1 scores after about 200 epochs, and the reduction of the loss function decreased. However, setting 400 epochs is mainly based on the following considerations:

(1) Global optimality exploration: although the model performance metrics are no longer significantly improved after 200 epochs, we note that the longer training period helps the model to jump out of the local optimal solution and search for possible better solutions, which may be beneficial to the model's generalization ability and stability even if the gain is small.

(2) Avoiding the risk of early stopping: stopping training early may lead to fluctuations in the model's performance on the validation set, and the choice of 400 epochs is intended to ensure that the model learns the diversity and complexity of the dataset adequately over a sufficiently long period of time, and to prevent potential performance improvement opportunities from being missed by ending the training too early.

(3) We considered the possible risk of overfitting, as well as the improvement in accuracy and precision. Ultimately, 400 epochs were chosen as the default training termination point, and doing so yielded better classification results in this study.

Combining the above reasons, we decided to set the number of training rounds to 400, which contributes to the robustness of the model even though the growth of the performance metrics slows down during the later stages of training. Meanwhile, we also noticed your question about the performance bottleneck at the late stage of training, and in our future work, we will consider introducing more refined training strategies, such as Early Stopping or other optimization methods, to save computational resources and avoid overfitting. Thank you again for your review and suggestions, and we will reasonably adjust and optimize the experimental scheme according to the actual situation.

**Comments 7:** *In the description of the evaluation metrics in Section 3.2.3, what are the validation set and test set mentioned? What the descriptions such as true positive and false negative represent? Please give more clear explanations.*

**Response 7:** We often thank you for your valuable suggestions on the evaluation metrics part of our study. In the revised Section 3.2.3, we have elaborated the concepts of validation and test sets and their roles in evaluating model performance:

In the field of machine learning and deep learning, datasets are usually categorized into three parts: training set, validation set, and test set. The training set is the part used to train the model to learn the intrinsic laws of the data; the validation set is used to adjust the model parameters during the training process and test the model generalization ability to determine the best model; and the test set is completely independent of the training process, and is only used to evaluate the final performance of the model on unknown data after the model training is completed to ensure that the evaluation results are fair and objective.

For the description of True Positive (TP) and False Negative (FN), we have explained these concepts and how they are calculated in detail in the " 3.2.3. Cloud Classification Evaluation Indicators " section of the revised draft. We have explained these concepts and their calculations in detail in the "3.2.3:

*"True Positive (TP) denotes the actual number of positive samples that the model correctly predicts as positive category (i.e. cloud category), which represents the number of real cloud images that the model successfully recognizes. False Positive (FP) denotes the number of samples that the model incorrectly predicts as positive category but actually belongs to the negative category (non-cloud category), which implies the number of cloud images that the model misidentifies. False*

*Negative (FN) denotes the number of samples that the model incorrectly predicted as a negative category but actually belonged to a positive category, which represents the number of cloud images that the model failed to identify. "*

By clarifying these concepts, we have ensured a clearer and more thorough presentation of the evaluation metrics section to facilitate the reader's understanding of how we quantitatively assessed the performance of the cloud classification model on the validation and test sets.

**Comments 8:** *In part 3.3, how to estimate the A value and how to reflect the adaptive process in the defogging algorithm? Do you mean that each type of cloud selects a different A value to achieve adaptive?*

**Response 8:** Thank you very much for your interest and guidance on the atmospheric light estimation and its application to adaptive defogging algorithms in Section 3.3 of the paper. In response to your questions, I will make a more explicit and detailed explanation in the revised text.

In our defogging algorithm, the A-value represents the global atmospheric light intensity, which plays a key role in the defogging effect of the whole image. In the dark-channel a priori algorithm proposed by Kaiming et al. (2009), we first calculate the minimum values of the three RGB color channels at each pixel point to construct a dark-channel image. Then, we use the non-zero minimum value in the dark-channel image to estimate the global atmospheric light intensity A. This value reflects the atmospheric light intensity in the scene that is not shaded by clouds and is directly illuminated by sunlight.

*"In the image enhancement algorithm, the atmospheric light value A directly impacts the intensity of dehazing. Thanks to the powerful cloud classification network, we design adaptive enhancement strategies after recognizing different cloud types. For relatively thin cirrus, excessive enhancement may filter them out, hence smaller A values are chosen to preserve details. "*

We have designed a strategy to dynamically adjust the A-value according to the morphological characteristics of the cloud body and the complexity of the lighting conditions. For different types of clouds, we will flexibly choose the most suitable A-value to optimize the de-fogging enhancement effect, which in turn improves the recognition accuracy of cloud edges and thinning regions. Thanks again for your review, we have added the above content in the revised manuscript to fully elaborate the estimation method of A-value and its dynamic adjustment process in the adaptive defogging algorithm.

**Comments 9:** *In figure 7, it can be seen that the contrast between cloud and clear sky is obviously enhanced after image enhancement. I am curious about whether the NRBR method is applied to the images before and after enhancement, and will the results be different? Please provide the cloud detection results of the NRBR method for the enhanced image and compare it with your new method.*

**Response 9:** Thank you for your interest in the comparative effect of image enhancement

before and after and the effectiveness of NRBR method applied on enhanced images. On the basis of our original research, we will compare the performance difference of NRBR method before and after image enhancement, firstly, we have processed the original image with adaptive enhancement, and then we apply the NRBR method and our new method of finite sector segmentation combining with K-means clustering to cloud detection of enhanced image respectively, and we get the results shown in Figure 1:

[Figure]

(a)

(b)

(c)

(d)

[Figure]

(e)

Figure 1. Cloud recognition effect comparison image. (a) Original cloud image; (b) The result after image enhancement; (c) Recognition effect of non enhanced images using NRBR method; (d) The recognition effect of the enhanced image using the NRBR method; (e) Finite sector segmentation k-means clustering results.

As shown in Figure Figure 1d, on the enhanced image, the NRBR method is somewhat improved in recognizing complex lighting conditions, especially the overexposed region (around the sun) and the cloud bottom details (on the right side of the image), but it is still not as good as our proposed novel method. The new method is able to recognize cloud boundaries (e.g., thin cloud on the left side and thicker cloud bottom on the right side of Figure 1e) more accurately and reduce confusion and misclassification when processing the enhanced image, especially in the sunlight-adjacent region with strong illumination, the edge of the thin cloud, and the bottom of the thicker cloud, where the new method shows a higher recognition accuracy.

**Comments 10:** *The algorithm considers using images from 9 : 00 to 16 : 00 in the day as training data, and the paper mentioned that the illumination has a great interference to the recognition results. What are the identification results before 9 : 00 and after 16 : 00 ? Are there any examples? In the cloud cover time series of Fig.8, you give the comparison between the proposed algorithm and the traditional algorithm. How do you calculate the improvement in accuracy of the new method compared to the traditional algorithm?*

**Response 10:** Thank you for your insights and specific questions about my research. In response to your questions, we add the following:

Regarding the results of image recognition for time periods other than 9:00 to 16:00 during the daytime, we did notice that image recognition outside of these two time points becomes more difficult and prone to misclassification due to the significant effect of lighting conditions on cloud recognition. In our initial experiments, we found that the overall whiteness of the sky could not distinguish between blue sky and white clouds in

the morning and evening hours due to the low sun angle, complex illumination and large intensity variations, leading to a decrease in the accuracy of cloud recognition. Moreover, due to the limitation of the sensor, the blue sky and clouds at night cannot be directly photographed by visible light, and need to be detected with the help of other data. We have shown the individual images before 9:00 and after 16:00 in Figure 2 below:

[Figure]

(a)                 (b)

(c)                 (d)

Figure 2. Cloud detection results before 9 a.m. and after 16 p.m. (a) Example of an all-sky image in the morning, just before 9 a.m., when the sun comes ups; (b) The cloud identification result of Fig. 2a; (c) Individual examples of all-sky images after 16 p.m.; (d) The cloud identification result of Fig. 2c.

Regarding the cloud coverage time series comparison shown in Fig. 8 of the original manuscript, we used the cloud amount calculation results of the two algorithms (i.e., the algorithm proposed in this study and the traditional NRBR algorithm) on the same set of image data and quantitatively analyzed them by comparing the accuracy of the cloud amount identification at 15:00 PM every day. We calculate the degree of match between the cloud amount identified by the new method and the actual naked-eye observation records in the image-enhanced data and compare it with the results obtained by the traditional NRBR method without enhancement. The improvement in cloud identification

accuracy of the new method is demonstrated by the statistically significant percentage improvement in accuracy and the reduction in the false positive rate under specific lighting adverse conditions such as the bottom of the cloud cover and overexposed regions around the sun.

**Comments 11:** *The classification accuracy of the algorithm is very high. But I'm interested in misclassified images. Can we know the specific reasons for these misclassifications?*

**Response 11:** Thank you very much for recognizing the classification accuracy of the algorithms in this paper, as well as expressing interest in the potential causes of misclassification. In the course of our research, we found that the identification of hybrid clouds is one of the main causes of misclassification. Hybrid clouds make it challenging for the algorithm to process such images due to their complex internal structure, the difficulty in defining the transition region of different cloud types, and the variation of cloud types under lighting conditions. In addition, complex lighting dynamics, especially in intense lighting as well as backlit scenes, may exacerbate the blurring of cloud body boundaries and features, further affecting the classification performance. We have examined the misclassified images output from the model one by one and found the following common problems:

(1) Complexity of hybrid cloud layers: hybrid clouds have diverse morphologies and complex internal structures, with both cumulus and laminar cloud elements, and this combination of features sometimes makes the model hesitant to make decision boundaries.

(2) Influence of lighting conditions: Under too strong or too weak lighting, the visual characteristics of cloud layers may be significantly distorted, leading to model difficulties in identification.

(3) Blurred edges of cloud bodies: the boundaries between cloud bodies and the sky are not clear in some images, which makes the model prone to confusion when performing segmentation and classification.

(4) Transitional clouds: Transitional patterns produced during the formation and dissipation of clouds are often difficult to categorize with typical cloud type features.

In future versions, we will strengthen our research on the cloud recognition problem under hybrid clouds and complex lighting conditions, and try to improve the accuracy of hybrid cloud recognition by improving the model architecture, optimizing the image preprocessing techniques, and introducing more hybrid cloud samples to train the model.

**Comments 12:** *In the third paragraph of the introduction, the sentence "Some studies train k-means models to swiftly cluster and recognize cloud and clear sky regions in whole sky images, improving cloud quantification speed and efficiency." lacks literature citation.*

**Response 12:** Thank you for your valuable comments on the paper.In the third paragraph of the introduction section, regarding the description related to the use of k-means model

for fast clustering and identification of clouds and clear sky regions in all-sky images to improve the speed and efficiency of cloud computation, you have pointed out that there is a lack of literature citation.We apologize for this and will fill in the academic basis for this section in the revised manuscript.

*"Krauz and other research teams have previously successfully analyzed all-sky images using the k-means clustering algorithm to quickly and efficiently delineate cloud cover and clear-sky regions, significantly improving the speed and efficiency of cloud quantification tasks (Krauz et al., 2020)."*

Thank you again for your professional guidance, which certainly helps us to improve the quality of our dissertation.

**Comments 13:** *When the "Yangbajing Comprehensive Atmospheric Observatory" appears for the first time, it is recommended to give specific latitude and longitude coordinates.*

**Response 13:** Thank you very much for your valuable suggestions. We have provided specific latitude and longitude coordinates for the first reference to the "Yangbajing Comprehensive Atmospheric Observatory" in the subsequent manuscript to enhance the reader's clarity in locating the site, with the following modifications:

*"The Yangbajing Total Atmosphere Observatory (90°33′E,30°05′N) is located next to the Qinghai-Tibet Highway and Qinghai-Tibet Railway, 90 kilometers northwest of Lhasa, Tibet, in an area with an average elevation of 4,300 meters."*

**Comments 14:** *In Section 2.3, pay attention to the number of samples, it should be 15 samples per day instead of 16.*

**Response 14:** Thank you very much for your careful review and correction of the sample size of data in this paper.In section 2.3, as you pointed out, we did incorrectly describe the study as collecting 16 samples per day. In response to your comment, we have checked the raw data and confirmed that 15 samples were actually collected each day. We have amended the relevant paragraph to read:

*"Considering that images during sunrise and sunset hours are susceptible to lighting conditions, we only select images between 9am and 16pm hours each day. Also, to reduce the correlation, only one image is selected every half hour, which results in 15 sample images per day."*

**Comments 15:** *In Part 2.2, what is CMOS? If it is an abbreviation, please give a full name.*

**Response 15:** Thank you very much for your valuable comments on my article. I apologize for the inconvenience caused to the readers by using the acronym CMOS instead of giving its full name in part 2.2 of the article.CMOS stands for Complementary Metal-Oxide-

Semiconductor, a technology widely used in the fabrication of digital image sensors, especially for capturing high-resolution sky images in the all-sky imaging systems covered in this paper. In the revised version, I will give a full explanation of the terminology when CMOS is first mentioned to ensure that the content is clear and understandable.The revised content is as follows:

*'This visual imaging device is equipped with a complementary metal oxide semiconductor (CMOS) image sensor system with an ultra wide angle fisheye lens design, which can regularly capture visible light spectrum images across the entire sky range; The integrated sun tracking system can accurately calculate and track the position of the sun in real-time, ensuring effective blocking of direct sunlight shining into the CMOS system, thereby protecting its sensitive photosensitive components from damage and significantly reducing the interference effect of white light around the sun on subsequent image processing.'*

**Comments 16:** *Figure 8 in "Likewise, the cloud cover recognition effect at the base of heavier clouds and the overexposed area surrounding the sun are greatly improved when the image is enhanced, as seen in Figure 8g,h." should be changed to Figure 7.*

**Response 16:** Thank you for meticulously reviewing and pointing out a figure citation error in the paper. You pointed out that when describing the improved cloud recognition effect in section 4.2 of the paper, the figure number cited should be Fig. 7 instead of Fig. 8. I apologize for this and have immediately corrected the manuscript. The revised presentation is as follows:

*"Similarly, after image enhancement processing, as shown in Figures 7d and 7h, the cloud recognition effect at the bottom of thicker cloud layers and overexposed areas around the sun was significantly improved compared to Figures 7c and 7g."*

In subsequent revisions, we will be more careful in verifying all references to charts and tables to ensure that the content is presented correctly.

**Comments 17:** *Some grammatical words in the article need to be checked carefully.*

**Response 17:** We take your comments about the grammar of the article very seriously, and we apologize for the poor grammar in the manuscript. We have spent a long time revising the manuscript, studying the language and readability, and involving professionals in correcting the language, correcting and optimizing the parts of the text that may contain grammatical errors, and trying to eliminate any barriers to reader comprehension.

We ask for your additional guidance and suggestions as you review the new revised manuscript. We firmly believe that after this comprehensive proofreading and revision, the linguistic quality of the article will be enhanced, thus better serving the dissemination and communication of scientific research. Thank you again for your attention and support to this article!

---

## Author Comment (AC4)

**Response to Reviewer 4 Comments**

Thank you for your letter commenting on our manuscript entitled "*Innovative Cloud Quantification: Deep Learning Classification and Finite Sector Clustering for Ground-Based All Sky Imaging*" (MS No.: egusphere-2024-678). These comments are valuable and very helpful for the revision and improvement of our paper. We have carefully studied and made corrections, and hope to get your approval. The main changes of the paper and the responses to the review comments are as follows.

==Comments 1:== *The description of the deep neural network architecture and the process of finite element segmentation and clustering is detailed and provides a clear understanding of the approach. However, the authors might consider including additional visual aids or flow diagrams that could further elucidate the step-by-step process, especially for readers who may not be as familiar with the technical aspects of neural networks and image segmentation.*

==Response 1:== Dear reviewers, thank you very much for your careful review and valuable suggestions on our paper. We fully agree with you that the description of deep neural network architecture, finite element segmentation and clustering process can be made more intuitive and understandable by adding visual aids, especially for those readers who are not familiar with neural network technology and image segmentation, and we will take the following steps to make the changes:

*Add a flowchart: an adaptive image segmentation flowchart (shown in Figure 1) is added to the paper, which clearly shows the whole cloud detection process from all-sky image preprocessing results, deep learning classification to finite sector segmentation with K-means clustering, which will help the readers to intuitively understand how the various steps are implemented in sequence and the logical relationship between them.*

Hopefully, this revision will improve the readability and comprehension of the article and ensure that the technical details are transparent and easy to grasp for all readers. Thank you again for your feedback, it is crucial for us to improve the quality of our paper, and we look forward to your further guidance on our revised manuscript.

[Figure]

Figure 1. Adaptive image segmentation process. (a) Image after preprocessing; (b) Sector segmentation based on cloud type; (c) Sector K-means clustering recognition; (d) Cloud recognition result.

**Comments 2:** *The authors have used an extensive dataset from the Yangbajing station in Tibet. It would be beneficial to discuss the representativeness of this dataset in the context of other geographic regions or climatic conditions. If the model's applicability is limited to regions similar to the dataset's origin, this limitation should be explicitly stated.*

**Response 2:** Dear reviewers, thank you very much for your valuable comments on our paper. You pointed out that we should discuss the representativeness of the extensive dataset originating from the Yangbajing station in Tibet in other geographic regions of the globe or under different climatic conditions, as well as clarifying whether the scope of model application is limited by the similarity of the regions from which the dataset originates. We fully agree with you that this is essential for assessing the generalizability and usefulness of the model. We have made the following changes and additions to the corresponding section of the paper:

*"Due to the limitations of single-site data in revealing broader patterns of cloud variability, we have decided to incorporate data from more diverse geographic locations and climate conditions in future research to enhance the model's applicability to various geographical environments and climate scenarios. We plan to establish a multi-site, cross-regional cloud cover and cloud type dataset, which, through integration and comparison of data from different locations, can not only validate and optimize the proposed cloud cover quantification method but also assess its applicability and accuracy in different climatic backgrounds. While this study only conducted instance verification at the Yangbajing station in Tibet, the method proposed exhibits strong scalability and universality."*

It is worth noting that we have designed our cloud quantization methods, including adaptive segmentation strategies, finite sector clustering, and illumination invariant image enhancement algorithms, to be flexible and scalable. Theoretically, with proper tuning and targeted training, these methods can be adapted to accommodate more diverse cloud cover and illumination conditions. However, the current research phase does have dataset geographical limitations, which we have explicitly pointed out in the paper to ensure scientific rigor in the interpretation and application of the results. We hope that the above additions express both a clear understanding of the limitations of the current study and a vision of the direction of future research.

**Comments 3:** *The paper presents impressive classification accuracy rates. However, it would be advantageous for the authors to include additional validation, possibly through a comparison with other state-of-the-art methods or by applying the framework to an independent dataset to verify its generalizability.*

**Response 3:** Thank you very much for recognizing our research and for your valuable suggestions. You mentioned that our paper demonstrated impressive classification accuracy, and also suggested that we further strengthen the validation, such as by comparing with other cutting-edge methods or applying the framework to independent datasets to verify its generalization ability. We fully agree with you that this will significantly enhance the persuasiveness and usefulness of our research results. Based on your suggestions, we plan to take the following steps to make additions and modifications:

*In our paper, indeed, we employed a comparative analysis using the YOLOv8 model against the BoMS[1] method from 2016 on the TCI dataset, which evidenced substantial improvements. We are acutely aware of the dynamism and rapid pace of technological advancements within the field, as exemplified by a recent study where a streamlined convolutional neural network based on MobileNet[2] architecture achieved an overall accuracy of up to 97.45% on comparable public datasets. Additionally, other state-of-the-art cloud classification networks such as CloudNet [3], Transformer-based models [4], and Combined convolutional network [5] have also shown commendable results.* However, due to the fact that this current work has not yet directly applied or conducted comprehensive comparative experiments with these cutting-edge algorithms on the same dataset, we were unable to provide a direct quantitative comparison in the present paper. Nonetheless, we wholeheartedly agree with your perspective and commit to incorporating these recent advancements in our future research agenda, thereby enabling a more thorough evaluation of the robustness and generalization capabilities of our YOLOv8 model architecture under complex meteorological conditions. *To better convey this information, we have included an additional table (See Table 1 below; this is Table 4 from Chapter 5 of the thesis) in the revised version, which presents a comparative overview of our model's performance against the latest techniques reported in the literature concerning cloud quantification metrics. This visual representation serves to clearly illustrate the relative strengths and weaknesses of various methods, thus validating the efficacy of our model and contributing to the provision of more accurate and efficient cloud quantification solutions for climate science research.* Once again, we thank you for your expert guidance and assure you that we will diligently incorporate your recommendations into the enhancement and refinement of our article's content.

**Table 1.** Comparison of this study with the latest technological approaches in the literature.

| Article | Dataset | Year | Model/Method | Accuracy(%) |
|---------|---------|------|--------------|-------------|
| Li et al. (2016) | TCI | 2016 | BoMS | 93.80 |
| Zhang et al. (2018) | CCSN | 2018 | CloudNet | 88.0 |
| Li et al. (2022) | ASGC
CCSN
GCD | 2022 | Transformer | 94.2
92.7
93.5 |
| Zhu et al. (2022) | MGCD
NRELCD | 2022 | Combined convolutional network | 90.0
95.6 |
| Fabel et al. (2022) | All sky images (Owned) | 2022 | Self-supervised learning | 95.2 |
| Gyasi et al. (2023) | CCSN | 2023 | Cloud-MobiNet | 97.45 |
| Ours | All sky images
TCI | 2023 | YOLOv8 | 98.19
98.31 |

[1] Li, Q.; Zhang, Z.; Lu, W.; Yang, J.; Ma, Y.; Yao, W. From pixels to patches: a cloud classification method based on a bag of micro-structures. Atmos. Meas. Tech. 2016, 9, 753-764. [CrossRef]

[2] Gyasi, E.K.; Swarnalatha, P. Cloud-MobiNet: An Abridged Mobile-Net Convolutional Neural Network Model for Ground-Based Cloud Classification. Atmosphere. 2023, 14. [CrossRef]

[3] Zhang, J.; Liu, P.; Zhang, F.; Song, Q. CloudNet: Ground-Based Cloud Classification With Deep Convolutional Neural Network. Geophys. Res. Lett. 2018, 45, 8665-8672. [CrossRef]

[4] Li, X.; Qiu, B.; Cao, G.; Wu, C.; Zhang, L. A Novel Method for Ground-Based Cloud Image Classification Using Transformer. Remote Sens. 2022, 14. [CrossRef]

[5] Zhu, W.; Chen, T.; Hou, B.; Bian, C.; Yu, A.; Chen, L.; Tang, M.; Zhu, Y. Classification of Ground-Based Cloud Images by Improved Combined Convolutional Network. Applied Sciences. 2022, 12. [CrossRef]

**Comments 4:** *While the paper addresses illumination dynamics and their impact on cloud quantification, it would be interesting to see a more in-depth analysis of how different lighting conditions, such as those during sunrise and sunset, affect the accuracy of cloud detection.*

**Response 4:** You mentioned that you would like to see a more in-depth analysis of the impact of different lighting conditions, especially the special lighting conditions at sunrise and sunset, on the accuracy of cloud detection, which is a very valuable research direction. We do notice that image recognition outside of these two time points becomes more difficult and prone to misclassification. At sunrise and sunset, the sun's angle is low and the light is oblique, resulting in a large change in the contrast between the light intensity on the ground and in the clouds, a change that may make the edges of the clouds blurry and increase the difficulty of distinguishing clouds from the background sky. For example, thin cloud layers and high altitude cirrus clouds may be difficult to recognize at dusk or dawn due to light scattering, affecting the accurate quantification of cloud cover. Due to sensor limitations, blue skies and clouds at night cannot be captured directly by visible light and need to be detected with other data. We have shown individual example recognition images for sunrise and night in Figure 2 below:

[Figure]

(a)        (b)

(c)        (d)

Figure 2 Cloud detection results at sunrise and night. (a) Example of full sky image at sunrise (b) Cloud recognition result in Figure a (c) Example of sky image at night (d) Cloud recognition result in Figure c

**Comments 5:** *The discussion on the scalability and versatility of the approach is promising. To bolster these claims, a section on potential modifications or adaptations required to apply this framework to different meteorological stations would be beneficial.*

**Response 5:** We strongly agree with the importance of your reference to the scalability and generalizability of the methodology in this paper, and your suggestion to add a discussion of potential adaptations or modifications needed to adapt the framework to different weather stations. With this in mind, we would like to add the following to the "5.2 Model scalability" section of the paper to strengthen our argument:

*"In this study, although the example validation is only carried out at the Yangbajing station in Tibet, the method is highly scalable and universal, and the constructed end-to-end cloud recognition framework has the ability of generalization, and can be adapted to the cloud morphology characteristics of other geographic locations after appropriate model fine-tuning in the following ways:*

*(a) The climate characteristics of weather stations in different geographic locations are very different, such as high humidity in the tropics, extreme low temperature in the polar regions, and complex terrain in mountainous regions, for which the image preprocessing module needs to be adjusted as follows, (1) Climate-adapted image preprocessing: introduce region-specific light models and adjust the atmospheric light parameter A value in the image enhancement algorithm to adapt to the changes in the light under different climatic conditions, e.g., for the high latitude regions, the processing intensity of the defogging algorithm is strengthened to cope with the frequent fog and low-light conditions in winter; (2) terrain influence compensation: for mountainous or urban environments, the original zenith angle cropping range is modified to ensure that cloud identification is not interfered by surrounding environmental factors.*

*(b) Differences in all-sky camera models, resolutions and installation locations used by weather stations require the following adjustments to the reading module, (1) Modify the lens parameters in the algorithm configuration file, such as the image cropping range, the image suffix (e.g., jpg, png, etc.), and the image resolution standard. (2) Adjust the common data interface to ensure that the system can seamlessly access different brands and models of cloud cameras and data recording equipment to achieve automatic loading and standardized processing of data.*

*(c) Considering the specific needs of different weather stations, the system can provide highly personalized configuration options: (1) Parameter number configuration template: Provide preset parameter templates to set the optimal identification parameters and algorithm configurations for different climatic regions (e.g., tropical rainforests, deserts, and poles) and the frequency of occurrence of cloud types. (2) Dynamic adjustment mechanism: Dynamically adjust the algorithm parameters, such as the K value of K-Means clustering and the threshold value of cloud type identification, according to the system operation status and identification accuracy, in order to optimize the identification effect."*

Thank you again for your review and guidance; we have made substantial improvements in the revised manuscript as suggested here and will fully reflect these improvements in future revisions of the paper.

**Comments 6:** *Ensure consistency in terminology, especially when referring to the various neural network components and cloud types, to avoid confusion.*

**Response 6:** Thank you very much for your valuable feedback, your correction on the consistency of terminology plays a key role in improving the quality of the paper. In response to your suggestion, we have carefully revised the terminology to ensure that all references to neural network components and cloud types are consistent throughout the paper, so as to avoid potential confusion, some of the changes are as follows:

*(1) The nomenclature of the four cloud types has been standardized, and the terms "cirrus", "clear sky", and "cumulus" are strictly used in the text, figures, and references. cumulus", and "stratus" are strictly used to ensure the accuracy and consistency of the terminology.*

*(2) For the adopted deep learning framework, we have clarified YOLOv8 as the unified title of the core algorithm in this paper, and maintained the consistency of this expression in all related discussions and descriptions, avoiding abbreviations or other variants that may cause confusion.*

*(3) The description of the internal structure of the YOLOv8 framework has been further calibrated to ensure that network components such as Darknet-53 and C2f modules are referred to with precise expressions that match the actual structure of the framework.*

We are confident that these revisions not only enhance the clarity and professionalism of the paper, but also enhance the reader's comprehension experience. Thank you again for your careful review and constructive comments, which have greatly contributed to the rigor of our research and the accuracy of our presentation. We look forward to your further guidance on the revised manuscript, and we are willing to make continuous improvements to meet the high standards of academic publication.

**Comments 7:** *The use of precision, recall, and F1-score is appropriate. Including additional statistical analyses, such as a confusion matrix, would provide a more comprehensive overview of the model's performance across all classes.*

**Response 7:** Thank you very much for your in-depth review of our study and your valuable suggestions. Your proposal to include a confusion matrix to complement the existing precision, recall, and F1 score evaluations is one that we fully agree with. Confusion matrix, as a powerful visualization tool, can indeed provide a comprehensive and nuanced view of the model's performance on all the categories, which helps to gain a deeper understanding of the classification correctness and error patterns among the categories, as shown in Figure 3 below for the validation set:

[Figure]

Figure 3. Confusion matrix results of the model on the validation set

Although we have not yet directly included the confusion matrix in the current submission, we value this constructive feedback from you and in the future, in further studies or extended versions of the paper, we plan to integrate the confusion matrix analysis to enhance the comprehensiveness of our model performance evaluation. This will not only help readers intuitively identify the strengths and weaknesses of the model across categories, but also facilitate effective comparisons with other research efforts, and we thank you again for your guidance in enhancing the rigor and transparency of our research.

**Comments 8:** *The authors have briefly mentioned future work in improving the model's adaptability to overexposed regions. Elaborating on potential avenues for future research, such as incorporating additional atmospheric parameters or exploring the effects of climate change on cloud dynamics, would be insightful.*

**Response 8:** Your insights about future work are pertinent and we deeply agree with them and have decided to further expand the discussion of future research directions by making the following additions and modifications to the Discussion section of the paper's conclusion:

*For overexposed regions: (1) plan to incorporate additional meteorological data, such as temperature, humidity, and wind speed, into our predictive models by combining these parameters with image data to refine our understanding of cloud formation dynamics and improve model accuracy under variable atmospheric conditions; (2) explore the temporal evolution of cloud patterns and their response to global warming trends, analyze historical and projected climate data to quantify how changes in temperature gradients, precipitation patterns, and atmospheric stability affect cloud morphology and distribution, and to develop models that can predict long-term changes in cloudiness, thereby contributing to climate prediction models; (3) To address the challenge of overexposure, we plan to investigate and implement state-of-the-art exposure correction algorithms, such as adaptive histogram equalization or high dynamic range (HDR) imaging, that can mitigate the effects of overexposure and thereby improve the accuracy of models under bright conditions.*

*effects, thereby improving the model's ability to accurately identify cloud features under bright illumination conditions; (4) combining ground-based imagery with satellite data and potentially other remote sensing techniques can provide complementary perspectives on cloud cover and dynamics, and integrating these different data sources may enhance our ability to comprehensively model cloud systems, especially in regions where ground-based observations alone may not be sufficient.*

**Comments 9:** *The cited literature it is currently poor, I suggest to the authors to cite relevant studies on cirrus clouds and their importance.*

**Response 9:** Thank you for your valuable comments on our paper, especially on the literature citation, we plan to enhance the literature support of the paper by including the following references:

[1] Gouveia, D. A., Barja, B., Barbosa, H. M. J., Seifert, P., Baars, H., Pauliquevis, T., and Artaxo, P.: Optical and geometrical properties of cirrus clouds in Amazonia derived from 1 year of ground-based lidar measurements, Atmos. Chem. Phys., 17, 3619-3636, 10.5194/acp-17-3619-2017, 2017. This study provides the one year of observational data on the optical and geometric properties of cirrus clouds in the Amazon region, which provides important information for understanding the role of cirrus clouds in the tropics.

[2] Marsing, A., Meerkötter, R., Heller, R., Kaufmann, S., Jurkat-Witschas, T., Krämer, M., Rolf, C., and Voigt, C.: Investigating the radiative effect of Arctic cirrus measured in situ during the winter 2015-2016, Atmos. Chem. Phys., 23, 587-609, 10.5194/acp-23-587-2023, 2023. The paper explores in detail the winter 2015-2016 field measurements of Arctic cirrus clouds, revealing their radiative effects, which are important for understanding the impact of polar cirrus clouds on the global energy balance.

[3] Shi, X. and Liu, X.: Effect of cloud-scale vertical velocity on the contribution of homogeneous nucleation to cirrus formation and radiative forcing , Geophys. Res. Lett., 43, 6588-6595, 10.1002/2016GL069531, 2016. This study focuses on the homogeneous nucleation process of cirrus cloud formation, especially the effect of intracloud vertical velocity on this process and the potential impact on radiative forcing, which provides a new perspective on the microphysical mechanism of cirrus cloud formation.

By citing this literature, we not only strengthen the scientific basis of the paper, but also enrich the discussion on the physical properties of cirrus clouds, their radiative effects, and their behavior under different regional and climatic conditions. We believe that these additions will significantly enhance the comprehensiveness and depth of the paper, and we look forward to your further review of the revised manuscript and welcome any additional feedback and suggestions.